# Lake Drainage in Permafrost Regions Produces Variable Plant Communities of High Biomass and Productivity

**DOI:** 10.3390/plants9070867

**Published:** 2020-07-08

**Authors:** Sergey Loiko, Nina Klimova, Darya Kuzmina, Oleg Pokrovsky

**Affiliations:** 1BIO-GEO-CLIM Laboratory, National Research Tomsk State University, Lenina St. 36, 634050 Tomsk, Russia; ninmilk@yandex.ru (N.K.); kuzmina.d.m.95@gmail.com (D.K.); oleg.pokrovsky@get.omp.eu (O.P.); 2Tomsk Oil and Gas Research and Design Institute (TomskNIPIneft), Prospect Mira 72, 634027 Tomsk, Russia; 3Institute of Monitoring of Climatic and Ecological Systems Siberian Branch of the Russian Academy of Sciences (IMCES SB RAS), Academichesky ave. 10/3, 634055 Tomsk, Russia; 4N. Laverov Federal Center for Integrated Arctic Research, Russian Academy of Sciences, Severnaya Dvina Embankment, 23, 163000 Arkhangelsk, Russia; 5Geosciences and Environment Toulouse, UMR 5563 CNRS, 14 Avenue Edouard Belin, 31400 Toulouse, France

**Keywords:** drained thermokarst lake, khasyrey, plant communities, NDVI, soil physical and chemical properties, western Siberia, low-Arctic tundra

## Abstract

Climate warming, increased precipitation, and permafrost thaw in the Arctic are accompanied by an increase in the frequency of full or partial drainage of thermokarst lakes. After lake drainage, highly productive plant communities on nutrient-rich sediments may develop, thus increasing the influencing greening trends of Arctic tundra. However, the magnitude and extent of this process remain poorly understood. Here we characterized plant succession and productivity along a chronosequence of eight drained thermokarst lakes (khasyreys), located in the low-Arctic tundra of the Western Siberian Lowland (WSL), the largest permafrost peatland in the world. Based on a combination of satellite imagery, archive mapping, and radiocarbon dating, we distinguished early (<50 years), mid (50–200 years), and late (200–2000 years) ecosystem stages depending on the age of drainage. In 48 sites within the different aged khasyreys, we measured plant phytomass and productivity, satellite-derived NDVImax, species composition, soil chemistry including nutrients, and plant elementary composition. The annual aboveground net primary productivity of the early and mid khasyrey ranged from 1134 and 660 g·m^−2^·y^−1^, which is two to nine times higher than that of the surrounding tundra. Late stages exhibited three to five times lower plant productivity and these ecosystems were distinctly different from early and mid-stages in terms of peat thickness and pools of soil nitrogen and potassium. We conclude that the main driving factor of the vegetation succession in the khasyreys is the accumulation of peat and the permafrost aggradation. The soil nutrient depletion occurs simultaneously with a decrease in the thickness of the active layer and an increase in the thickness of the peat. The early and mid khasyreys may provide a substantial contribution to the observed greening of the WSL low-Arctic tundra.

## 1. Introduction

Ongoing climate changes in the Arctic are leading to longer, warmer growing seasons, greater nutrient availability and precipitation, enhancing plant productivity and thermokarst activity [1,2,3,4,5]. The effect of increased productivity is known as tundra greening and has been confirmed both by satellite-derived Normalized Difference Vegetation Index (NDVI) and in-situ research [6,7]. The greening is not only caused by increased vegetation activity but more importantly by widespread colonization of previously unvegetated drained lake basins [8]. The opposite effect is associated with a decrease in the NDVI (i.e., browning) and usually indicates reduced vegetation growth [6]. Greening and browning are observed in various subarctic regions, although trends are strongly variable and heterogenous over space and time [9]. For example, only 18% of the total area of the northern West Siberian Lowland (WSL), the largest province of permafrost peatlands in the world, had statistically significant changes in productivity, with 8.4% increasing (greening) and 9.6% decreasing (browning) [10,11]. Between 1982 and 2003, against the background of an increase in terrestrial greening, an increase in browning also occurred, but the two processes are separated in space [12].

In the low-Arctic tundra of northern WSL, the average annual air and permafrost temperature and precipitation have increased over the past decades [13,14,15,16,17]. As a result, the active layer thickness (ALT) (seasonally unfrozen layer of the permafrost) has also increased [18,19]. The changes in climatic and thermal regimes has led to increased frequency of thermokarst events [20], such as a full or partial drainage of thermokarst lakes [21,22]. The increase in frequency of thermokarst lake drainage was noted in different regions of the Arctic, including part of the permafrost zone of Western Siberia, located in the mid-Arctic tundra [23,24,25,26], North eastern Siberia and the Far East [24,27,28], Canada [29,30,31,32] and Alaska [33,34,35,36]. In some areas lakes drainage has occurred simultaneously with an increase in the area of remaining lakes (for example, in Central Yakutia), which is caused by an increase in precipitation [28]. Increasing the lake size has a positive effect on the trend of browning. In the WSL, lake drainage occurs due to the thawing of the permafrost and the formation of soil subsidences in the valley through which lake water flows. The soil stability decreases, which causes a deepening of the flow and the formation of erosion channels, through which the lake water is discharged [21]. The thawed ground is washed away by a stream, which leads to a decrease in the water table of the lake [37,38]. As a result, lacustrine sediments become exposed. Pioneer plant species colonize the relatively warm and nutrient-rich sediments, thus initiating the succession of vegetation and soil [22].

The mineral and peat permafrost soils of the Subarctic accumulate nutrients in the permafrost layer near the interface with active layer [39,40]. Thermokarst activity and active layer deepening exert strong control on the availability of nutrients in soils [41,42,43,44]. An increase in the average annual air temperature increases the intensity of organic matter decomposition, thus further enriching the ecosystem in nutrients [45]. Furthermore, deepening ALT increases potential for export of inorganic nitrogen (N) from permafrost-influenced soils to the aquatic systems [46,47]. The hydrological export of carbon and nutrient strongly increases due to collapsing permafrost of upland and polygonal bogs, due to the leaching of nutrients, previously preserved in permafrost [48,49]. Plants with deep root systems, such as sedges which are the dominant pioneer species, are particularly sensitive to the increase in nutrient cycling [50].

The drained basins of thermokarst lakes, called ‘khasyreys’ in the WSL, are influenced by a superposition of two major factors of tundra greening. The first is a decrease in open water area as a result of lake drainage, followed by an increase in nutrient input from catchments. The second factor is nutrient-rich sediments which favor the development of productive plant communities. The drainage of thermokarst lakes triggers a post-drainage succession. The initial post-drainage phase is characterized by gradual ground cooling associated with formation of permafrost [51] and the growth of segregated near-surface ground ice [52]. Permafrost aggradation occurs differently in different parts of the basin. The temperature of permafrost and ALT increases after the thickening of snow. In the low-Arctic tundra, the snow cover is thicker in micro-depression and on sites with shrubs. The coldest soils are located at micro-elevations (the highest point) [53,54]. Ground ice accumulation at micro-elevations causes vertical movements of the surface. Local thaw subsidence leads a lower surface elevation. These processes create micro-mosaic of wetter and drier areas in basins [52,55,56]. After several decades, newly formed permafrost is present under frost mounds with peat sediment, but it is absent under wet marshy meadows [51]. The variation of plant communities within the basin will influence the values of NDVI observed from satellites. Therefore, in order to validate the NDVI data, extensive ground-based observations of post-drainage ecosystems are needed, in accordance with their microtopographic settings. Furthermore, comparing NDVI trends derived from satellite remote sensing with the results of ground-based observations will improve our understanding of the mechanisms of greening and browning processes [7]. This knowledge can be incorporated into model predictions of ecosystem development under various climate change scenario.

The sequence of vegetation, carbon fluxes, and the effect of nutrient availability on plant phenology have been extensively studied in drained lake basins in Canada and Alaska [1,30,57,58,59]. In the boreal zone, drained lake basins are common in Central Yakutia, where they are called ‘alases’. Alases occupy up to 10–15% of the total area of permafrost plains [60,61,62,63]. However, due to climatic and geochemical features, Yakutian alases are very different from the lake drainage basins of the WSL peatlands. In Western Siberia, khasyrey vegetation was previously studied in the forest-tundra and northern part of the forest zone [22,64], as well as in the tundra zone of Yamal [65]. However, a combined approach of landscape microtopography, plant ecology and biogeochemistry in the khasyreys of the southern part of the WSL tundra zone has never been attempted.

The first purpose of this work is to provide new insights into the vegetation of khasyreys in the WSL low-Arctic tundra. In particular, we characterize the species composition, ecological structure, phytomass and productivity of vegetation, elemental composition, and succession of the species. The second purpose to study the relationship between the vegetation parameters and the microtopography of khasyreys. The obtained results allow better understanding of the role of thermokarst lake drainage in plant diversity and biomass in the poorly studied region of permafrost peatlands of the Eurasian Subarctic via: (i) identifying the drivers of plant succession in khasyreys, and (ii) quantifying possible contribution of khasyreys to the greening of tundra.

## 2. Results

### 2.1. Microtopography and Soil Setting

The microtopography, soil and vegetation parameters are compiled in Table A2 and illustrated in Figure 1. The microtopography of young khasyreys follows that of the lake bottom and exhibits gentle slope from the shores to the center. On this slope, there are small mounds and depressions with an amplitutde of ca. 30 cm, linked to wave ripples of the former lake bottom. Soils are represented by Fluvisols in the central part and peat shores, and Gleysols, which are encountered in the vicinity of mineral shores, having sand layers. The ALT is 153 ± 58 cm but locally decreases to 50–80 cm in sites of peat deposits in the sediments (Table 1). The average litter thickness is 1.3 ± 0.6 cm (Table 1, Figure 1 and Figure 2). Strong horizontal variation of sediments thickness, humidity of soils and snow depth lead to laterally uneven aggradation of permafrost. Over the first decade in early khasyreys, the frost mound are formed in sites containing redeposited peat in the upper part of sediment profile.

These mounds achieve the maximal heights of 0.8–1.2 m after several decades, in mid khasyreys. At this stage, the heterogeneity of relief and environmental conditions is highly pronounced (Table A2, Figure 1). In average, the ALT is 114 ± 71 cm (Table 1, Figure 1 and Figure 2). On frozen mounds, the ALT decreases to 0.5 m whereas in depressions, the ALT remains ≥ 1.5 m. The relative height of mounds is 0.8 to 1.0 m. The most common soils are the Cryosols (with cryoturbation) and Gleysols, whereas the Fluvisols are developed along streams. The litter thickness is 6.2 ± 4.3 cm (Table 1, Figure 1 and Figure 2).

The khasyreys of the late stage exhibit minimal lateral heterogeneity. The peat thickness is 28.5 ± 10.7 cm (Table 1). The permafrost aggradation occurs throughout the entire basin. The ALT never exceeds 100 cm and equals on average 69 ± 65 cm (Table 1, Table A2; Figure 1 and Figure 2). The Histoisols and Gleysols dominate, whereas at the frozen mounds the Cryosols are abundant. The mounds of maximal size (350 × 130 m) and 0.5 to 0.8 m height are present in drained basin La3. The slopes of mounds are always gentler compared to those of the Mid stages. 

The median values of soil fertility parameters vary systematically between stages (Table 1, Figure 2). The pH of the soil solutions of the root layer is slightly acidic, amounting to (mean ± standard deviation) 6.3 ± 0.7 and 6.5 ± 0.7 in the early and mid, respectively. In the late khasyreys, the soil waters become very acidic (pH = 5.1 ± 0.7). The Specific conductivity, which integrally characterizes the availability of mineral nutrition elements, in the early and mid khasyreys is 208 ± 205 and 171 ± 170 µS·cm^−1^, respectively. In the late khasyreys, this parameter strongly decreases to 25 ± 5 µS·cm^−1^. The pools of labile phosphorus in the 0–30 cm layer are the largest in the soils of the early and mid khasyreys and strongly decrease during the late stage (13.8 ± 8.4, 13.3 ± 6.2 and 4.0 ± 2.2 g·m^−2^, respectively). The labile potassium (K) pool is the largest in the khasyreys of the mid stage at 35.8 ± 18 g·m^−2^, and it decreases to 20.8 ± 10.1 g·m^−2^ in the early stage. The difference between these stages is statistically significant (*p* = 0.0215). The minimum pool of labile K is in the soils of the late stage (9.6 ± 7.2 g·m^−2^). The mineral nitrogen exhibits the largest pool in soils of the mid stage (7.4 ± 5.5 g·m^−2^), whereas a minimal pool of N_mineral_ is observed in soils of the early successional stage of khasyreys (2.5 ± 1.8 g·m^−2^). In soils of the late stage, the N pool is 5.4 ± 1.6 g·m^−2^. Therefore, the ecotopes of the mid successional stage of khasyreys exhibit the highest fertility.

According to non-parametric H-criterion Kruskal Wallis for un-paired data (at *p* < 0.05), all parameters of soils and vegetation are different between succession stages (Table A3A). A pairwise comparison of stages revealed most significant differences in all parameters between Early and Late stages (Table A3B). The early and mid-stages are different in terms of litter/peat thickness (*p* = 0.001), N_min_ (*p* = 0.0255) and K_labile_ (*p* = 0.0215); the pools of nutrients are higher at the mid stage. The mid and late stage are different by their peat thickness (*p* = 0.003), soil density (*p* = 0.001), pH soil water (*p* = 0.003), Sp. cond. (*p* = 0.008), P-PO_4_ (*p* = 0.004), and K_labile_ (*p* = 0.002). Analysis of microtopography and soil types demonstrates that all three stages are sizably different, especially the early and late stage, which do not show similarity in none of the considered parameters.

### 2.2. Plant Communities and Successional Stages of Drained Lakes

The early successional stage is represented by plant communities with a dense herb layer and very sparse moss layer (Figure 3; Table A2). The dominant species are sedges, grasses, and in a lesser degree, cotton grasses (Figure 1 and Figure 4). All of them are common for floodplain meadows. Most plant species are hydrophilic (hydromesophytes and subhydrophytes) and flood tolerant (helophytes). These mesotrophic and mesoeutrophic species require high fertility.

In plant communities of the mid-successional stage, the cover of herb layer decreases, and the cover of moss layer increases in comparison with the ecosystems of the early stage (Figure 1). The composition of species is close to that in the communities of the early stage. However, the relative abundance of species which require soil trophicity decreases and they become less mesoeutrophs, more mesotrophs and even mesooligotrophs (Table A2). On the frost mounds with thicker litter (6–11 cm), the species which are common for tundra communities are abundant (*Betula nana*, *Vaccinium uliginosum*, *Dicranum elongatum* and others, see Figure 5a). The structure of plant communities of the mid successional stage varies among different ecotopes. On the flat bottom areas, the hydrophilic species (hydromesophytes and subhydrophytes) prevail, like in the early-stage communities (Figure 5b,c). However, on the frost mounds, there is high proportion of mesophytes—species, which avoid flooding: reedgrass, horsetail, or willows (Figure 5d). Sphagnum mosses of small abundance appear on some wet areas (Figure 6). Over the vegetation succession in wet ecotopes, the abundance of *Carex aquatilis* increases in the transition from the early to the mid stage, and the abundance of *C. rostrata* decreases.

In the khasyreys of the late successional stage, the plant communities are represented by typical tundra species. The total cover of the dwarf shrub-herb layer varies greatly. On the flat bottom areas, herbs (sedges and cotton grasses) form a sparse layer with a small admixture of dwarf shrubs. On the frost mounds, dwarf shrubs (*Ledum palustre*, *Betula nana*) form dense thickets (Figure 7a). The moss-lichen layer is continuous. Among the mosses, sphagnum prevail (dominated by *Sphagnum balticum*), both on the fens (Figure 7b,c) and on the frost mounds (Figure 1). Only on the highest and widest mounds, lichens prevail over the sphagnum. Late-stage plant communities differ from early- and mid-stage communities by the ratio of ecological groups of species in terms of soil trophicity. In the plant communities of the late successional stage, the dominant species are mesooligotrophs, whereas the share of mesotrophs decreases. Mesoeutrophic and mesotrophic sedge and cotton grass species (as at the early and mid-stage) are abundant only locally on the shores of residual ponds with deep ALT (>1.5 m) (Figure 7d). In terms of soil moisture, the structure of the late-stage plant communities varies in different ecotopes. Hydrophilic species (hydromesophytes and subhydrophytes) prevail on the flat bottom areas (fens), while mesophytes and hydromesophytes dominate on the frost mounds.

The phytoindication of ecotopes confirm these results on plant communities (Figure A1). In relation to the soil moisture, the ecotopes of the flat bottom areas are very close at all successional stages and correspond to wet conditions. However, on the frost mounds, which occur in areas of the mid and late stages, the soil moisture of ecotopes is lower. In relation to the soil trophicity, the ecotopes of the early and mid-stages are richer in nutrients than the ecotopes of the late stage, both the flat areas and the frost mounds. The ecotopes of the early stage are slightly richer than those of the mid stage; both correspond to mesotrophic conditions. On the frost mounds of the mid stage with relatively thicker litter, the soil richness is lower and comparable to that of the late stage. Late stage ecotopes are the most depleted in nutrients and correspond to meso-oligotrophic conditions; only on the shores of residual ponds with greater ALT (>1.5m) the conditions are mesotrophic.

### 2.3. Biological Productivity of the Vegetation in Khasyreys

The biological productivity of studied khasyreys were highly variable and decreased over one order of magnitude from Early to Late stages (Table 1). According to the H-criterion of Kruskal Wallis, the ANPP and NDVImax were significantly different between the three stages (Table A3B) with *p* = 0.0005 and 0.0001, respectively. Only in one case, during comparison of Early and Mid stage, the Mann-Whitney U-test did not reveal significant differences in the ANPP. However, a pairwise comparison of different stages according to the NDVImax always demonstrated significant differences between stages (*p* ≤ 0.003, Table A3B). Such inconsistency in behavior of ANPP and NDVImax may be due to their non-functional relationship, that can be approximated by an exponential dependence with R^2^ = 0.62 (Figure 8a).

The ANPP of vegetation of early and mid-successional stages demonstrates a very high productivity (Figure 9). The average ANPP and NDVImax of early stage plant communities are 1134 ± 645 g·m^−2^·y^−1^ and 0.68 ± 0.03, respectively (Table 1). These are the largest values among the three stages. The ANPP values of these stages (660–1134 g·m^−2^·y^−1^) are much higher than those of the WSL low-Arctic tundra (109 g·m^−2^·y^−1^, see Figure 9). The differences in the productivity of meadows composed of different hydrophilic species are largely determined by the fertility of soils of the corresponding ecotopes. The greatest values of plant productivity are recorded in meadows with a predominance of *Arctophila fulva*, confined to sites with highly fertile soils. The average productivity for these meadows was 1753 ± 824 g·m^−2^·y^−1^, and the highest value was 2538 g·m^−2^·y^−1^. The maximal value of productivity for the meadows dominated by *Carex aquatilis* was 2499 g·m^−2^·y^−1^, close to the maximum of arctophilic meadows. This unique value was noted within the residual pond shore near a collapsing polygonal bog, which is known to provide an additional amount of nutrients (i.e., Loiko et al. [66]). In contrast, the minimal value (454 g·m^−2^·y^−1^) was obtained in the sedge meadow located on a sandbank.

The differences in the ANPP among successions are governed by identity of species in plant communities, notably the degree of moss cover. For example, the ANPP of mid-succession of the arctophilic meadow decreased to 381 g m^−2^·y^−1^ due to an increase in brown moss cover (up to 65%) in the riparian zone of the stream (Table A2). The average ANPP and NDVImax of mid stage plant communities are 660 ± 292 g·m^−2^·y^−1^ and 0.64 ± 0.02, respectively (Table 1). The maximum value of the productivity of meadow with *Calamagrostis langsdorffii* (1438 g·m^−2^·y^−1^) was measured at the top of the frost mound in a community with <1% of moss cover. The minimum value of 223 g·m^−2^·y^−1^ occurred in the wet meadow of thermokarst subsidence, where the herb layer was not so dense (15–20%), and the moss cover was high (80–90%).

Plant communities of the late successional stage were characterized by minimal ANPP and NDVImax, which was significantly (*p* < 0.002) lower than that in the communities of the early and mid-stages and was close to the productivity of typical phytocoenoses of the low-Arctic tundra (Figure 9). The average productivity of the herbs, without mosses, in the sedge-cotton grass-sphagnum fen was 79 ± 22 g·m^−2^·y^−1^. The average total ANPP and NDVImax of late stage plant communities is 261 ± 77 g·m^−2^·y^−1^ and 0.54 ± 0.07, respectively (Table 1). Such low values are linked to thick peat layer (Figure 8b), which prevents vascular plants from absorbing nutrients from the underlying lake sediment. 

For several ecosystems, the vertical distribution of the live root mass was measured within the 0–30 cm soil layer, with a spatial resolution of 5 cm (Figure A3). The pattern of distribution is affected by the age of the drainage, soil moisture, and the ALT. The ratio of aboveground and underground phytomass of the studied meadows (Figure 10) show that three communities, where aboveground phytomass is higher than underground are formed on the most fertile soils of early successional stage (Figure 10a,b,f). For the plant communities of the mid stage, an excess of underground over aboveground phytomass is revealed. The highest ratio of the underground to aboveground biomass is observed in the horsetail (*Equisetum fluviatile*) meadow (Figure 10e), which occupy the ecotope with the lowest soil fertility.

### 2.4. PCA of Soil Trophicity and Vegetation Productivity

The PCA of the full dataset yielded two possible factors contributing to the observed variations in trophic soil parameters and vegetation productivity (i.e., 45% and 23%, respectively, Figure 11a). The first factor controlled all parameters, except mineral nitrogen, whereas the second factor was defined by ANPP, NDVImax, K_labile_, P-PO_4_, and N_mineral_. Factor 1 is most strongly correlated with soil density (R = −0.90), pHw (R = −0.82), peat thickness (R = 0.87) and NDVImax (R = –0.72). A medium-strength negative correlation was found between the ANPP and factor 1 (R = −0.37). Factor 2 has a strong negative relationship with nutrient pools in the soil, especially with N mineral (R = −0.79). Factor 2 has a positive correlation with ANPP (R = 0.67) and NDVImax (R = 0.47). The correlation matrix showed that most of the considered parameters determining soil fertility were mutually correlated (Appendix B, Table A4). Moreover, the ANPP has a pronounced correlation with the peat thickness (R = −0.52), and the NDVImax correlates well with soil density (R = 0.60).

The distribution of all studied sites in the space of two factors (Figure 11b) demonstrates that late stage forms a cluster along the positive values of factor 1. Most of the points in the early and mid stages are associated with negative values of factor 2 and not clustered.

### 2.5. The Elemental Composition of the Dominant Plant Species of Khasyreys

Chemical analyses of the aboveground mass of dominant plant species demonstrate that plant species are grouped according to successional stages (Table A5). Among the major elements, the lowest values are measured for Na, which strongly distinguishes khasyrey plants from previously studied macrophytes of forest-tundra and tundra lakes, where Na concentrations are much higher [69]. The highest Mg concentrations are observed in *Calamagrostis langsdorffii* at the sites of the mid stage, where they exceed the concentrations at the sites of the early stage. The minimal value of Mg concentration is noted in *Carex aquatilis*. The highest N concentrations and the minimal C:N ratios are characteristic of plants at the early stage. There is no distinct pattern in the distribution of P and Si among successional stages. The concentration of K is higher in plants of the early stage. Calcium concentrations are also highest in *Calamagrostis langsdorffii* growing on toeslope of the mid khasyrey. Aluminium concentrations are low in plants of the sites of the early and mid-stages, but they increase in sphagnum mosses of the late stage. Manganese concentrations are highest in *Carex aquatilis* at the mid stage. Iron concentrations are highest in sphagnum mosses, which may be due to the deposition of Fe(III) amorphous hydroxides on the surface of these photosynthesizing mosses in response to a rise in pH and O_2_ [69]. The ash content is the highest in *Calamagrostis langsdorffii*, notably at the toeslope of frost mound of mid khasyrey (2.8 ± 0.1%). The maximum values of pools of labile K and P-PO_4_ are confined to this site (Table A2).

The availability of nutrients in the ecosystem is reflected by their total pool in soil and vegetation. In this work, the chemical composition of the roots was not measured; therefore, the total pool was estimated only for the aboveground phytomass. The ratio of the P pools (phytomass + soil, per one taken a pool in late ecosystems) in the ecosystems of three stages of 7.3 (1st stage): 2.8 (2nd stage): 1 (3rd stage) reflects a much stronger decrease of P, compared to the ratio of N (3.0 (1st stage): 1.8 (2nd stage): 1 (3rd stage)) and K (2.8 (1st stage): 4.8 (2nd stage): 1 (3rd stage).

## 3. Discussion

### 3.1. Drivers of Plant Succession in Khasyreys

Development of vegetation is primarily controlled by the spatial heterogeneity of the lake sediments and the relief of the bottom. The most important factors are: (1) the heterogeneity of the physical properties of the substrate, (2) concentrations of mineral nutrients elements and particle-size distribution of the sediment, (3) soil moisture, and (4) duration of flooding. In addition, the permafrost aggradation leads to a differentiation of the microtopography, which, in turn, determines the distribution of the vegetation in the khasyreys.

Soil fertility is higher in early and mid-stages which favors the formation of plant communities dominated by *Arctophila fulva*, *Carex aquatilis* and *Carex rostrata* [42]. The colonization of *Arctophila fulva* in the lake basin, just after drainage, is due to the ability of this species to spread rapidly, with perennial creeping roots, thus producing large phytomass. This species withstands summer water flooding of 20–40 cm during the growing season. However, when the depth of flooding decreases, *Carex* sp. efficiently compete with *A. fulva*. A slow increase in the abundance of *C. aquatilis* in plant communities of overgrowing lake basins is also noted for the Northern Arctic plain of Alaska [57,70].

The microtopography associated with permafrost aggradation in khasyreys is closely linked to soil conditions (Figure 1). Already at the early stage, small (several dozens of centimeters) frost mounds colonized by willow are formed. These mounds are most likely raised in winter due to the formation of local ice lenses within the peat layers of lake sediments. In winter, growing willow bushes accumulate snow, which leads to the permafrost thawing and subsidence. This further leads to secondary pond formation, in which willows die due to over-wetting (Figure A4). Over the course of 10 years, willow bushes transition from the primary overgrowth of the khasyrey bottom to dying out due to flooding. Another important factor contributing to the aggradation of permafrost and the change in soils and vegetation over time is the accumulation of newly produced plant litter and its gradual transformation into a peat horizon (Figure 1 and Figure A5). Under the thick litter, permafrost is usually preserved for a longer time, which can later lead to the formation of an elongated frost mound along the shore ledge. After decreased frequency and duration of spring flooding, the grassy litter of *Arctophila fulva* remains on site of its accumulation and turns into a peat litter.

The colonization of sphagnum mosses at a late successional stage enhances the accumulation of peat and increases the thickness of organogenic horizons (Figure 1). A decrease in the active layer thickness suppresses the growth of willow bushes and graminoids, which are replaced by dwarf shrubs and mosses. This further decreases the snow accumulation in winter and consequently leads to less thawing. Ultimately, perhaps after a couple of thousand years, a polygonal bog will form in the khasyreys. This is consistent with the fact that the soil waters of the late khasyreys have pH and specific conductivity similar to those previously obtained [71] in the fen of the frozen polygonal bogs of the Taz tundra.

The PCA results demonstrate that the plant productivity is mostly controlled by the peat thickness (Figure 11). We consider the peat accumulation as a process that controls the changes in all other parameters used in the PCA model (except for Nmineral pool). In this case, a decrease in the Sp. cond. and the pool of labile P and K in the course of succession can be explained by the disconnection of rooting zone from nutrient-rich lake sediments occurring during the peat thickening. The ANPP via Factor 1 is positively correlated with labile P and K. However, via factor 2, the ANPP is negatively correlated to pools of the mineral N and, to a lesser extent, to labile P and K. This rather unexpected result can be explained by the rapid uptake of nutrients from part of the soils by actively growing biomass. This means that the productivity of individual ecosystems can be limited by nutrient availability. At the same time, the nitrogen pool in soils of the late successional stages is high, although the productivity of these ecosystems is low. This apparent contradiction can be explained as follows. The ratio of the N content in the aboveground phytomass (N_plant_) to the N mineral content in the 0–30 cm soil layer (N_soil_) is equal to an average of 5.7 for the site of the early successional stage, 1.1 for the site of the mid stage, and 0.5 for the site of the late stage. In other words, at the early and mid-successional stage, the plants strongly uptake the pool of N (N_plant_:N_soi_l > 1), whereas at the late stage, the soil N pool is not completely exhausted (N_plant_:N_soi_l < 1). The N:P ratio in the phytomass of WSL khasyrey ecosystems ranges from 3.7 to 12.2. The threshold of N vs. P limitation (N:P < 13.5, [72]) suggests that the plant communities are N-limited rather than P-limited. Overall, although the evolution of nutrient pools differs between elements, their total content in the ecosystem systematically decreases from the first to the late stage of khasyrey development, which reflects an increase in the oligotrophicity of the ecosystem.

### 3.2. Possible Contribution of Khasyreys to the Greening of the Tundra

Results of this study strongly confirm the recent observations that the emergence of highly productive ecosystems of khasyreys can provide a significant contribution to the greening of the tundra [5,8]. The plant communities of the early and mid-successional stages are well distinguished by their respective NDVI values (Figure 9; Table A3B). The NDVI of the first two stages is higher than that of both the phytocoenoses of the late successional stage and the surrounding polygonal tundra. This is consistent with our observations that, at the early successional stage, the productivity of plant communities is slightly higher than that at the mid successional stage. Among all tundra ecosystems, only the willow and reed grass communities growing in small ravines are comparable in their productivity to communities of the early khasyreys. According to the NDVI values, the ecosystems of the late khasyreys are also drastically different from the background tundra (Figure 9). We suggest that such a low productivity of communities at the late successional stage of khasyrey is due to thick peat layer, protecting nutrient-rich lake sediment from plant roots as demonstrated by a strong negative correlation between the NDVI and peat thickness (r^2^ = 0.71, see Figure 8b). This is consistent with observations in the Seward Peninsula lake basins where significant negative correlation (r^2^ = 0.63) between these parameters was reported [73].

Results obtained in this study allow better understanding of possible reasons of the WSL greening. In the northern part of the Pur-Taz interfluve, the proportion of “greened” ecosystems reaches one of the highest values in the cryolithozone of Western Siberia [10,11]. Miles and Esau [10] note that they did not have enough auxiliary data to reveal the nature of the trend for an increase in NDVI between 65° and 70° N latitude. It is known that various disturbances in the tundra, including thermokarst, can increase the vegetation productivity and therefore the NDVI [8,74]. Thus, our data allow identifying one component of the landscape—khasyreys—which can contribute to this increase in the NDVI. Miles and Esau [10] found that the effect of greening is characteristic of 8.4% of the WSL. It turns out that the greatest contribution to this effect is provided by khasyreys of the early and mid- successional stages, whose area is 5% [75].

### 3.3. Comparison with Other Regions and Ecosystems and Overall Significance of WSL Khasyreys

The basins of drained thermokarst lakes are common for the Subarctic plains of Northern Eurasia. The most typical and widely abundant are the alases of Central Yakutia [60,61,62,63]. While the alases are also formed in the continuous permafrost zone, the peat accumulation rate is low, and the thickness of peat in the alas ecosystems of wet and mid meadows does not exceed 10–15 cm [61], compared to the 20–40 cm in the khasyreys of the Pur-Taz interfluve studied in this work. This is probably due to the more arid climate of Yakutia, where the potential evaporation in summer is 4–10 times higher than the amount of precipitation [76]. This greatly limits the paludification and peat formation and contributes to the long and stable existence of meadows, some of which turn into lakes during wet years [63,77]. The composition of plant species in the drained lake basins of the northern territories (forest-tundra, tundra) differs significantly from that of the alases in Central Yakutia (forest zone, middle taiga subzone). This difference is caused by drier climates and the much higher soil salinity of alases [61,78].

It is interesting that a similar succession scenario was suggested for the discontinuous permafrost zone of Western Siberia [22,64] and NE part of European tundra [19,79]. However, due to the position of lake basins in forest and forest-tundra biomes, the final successional stages of the discontinuous and sporadic permafrost zone are different from those observed in the tundra zone. For example, for the khasyreys of northern taiga of the WSL (permafrost-free khasyreys), the ridge-hollow bog complex is considered as the final stage of succession [22,64].

The drained lake basins in the Arctic are not always subjected to colonization (overgrowth) by herbs. In the Old Crow Flats (Yucon, Alaska), the drained lake basins become covered with willow thickets [30]. In the northern taiga of the north of European Russia, the role of the willow in plant communities is also significant [19,79]. In contrast, in the Northern Arctic plain of Alaska, as well as on the Pur-Taz interfluve, hydrophilic grasses, including *Dupontia fisheri* and *Arctophila fulva*, together with *Eriophorum scheuchzeri*, predominate in the early stages of succession [58,60]. In the artificially drained Lake Illisarvik, within the Mackenzie Delta region (western Arctic coast, Canada) 40 years later, willows and birches occupied about a third of the area of the drained basin [80]. In general, the willow cover is higher in the khasyreys of the forest zone, compared to the khasyreys of the low-Arctic tundra.

The plant communities of the early and mid-successional stages of khasyreys have much in common in terms of species composition and structure with floodplain communities. Indeed, similar phytocoenoses have been described in the floodplain of the Ob River and the low reaches of the Pur, Taz, and Nadym rivers [81]. Moist ecotopes occupy the meadows of *Arctophila fulva*, which are replaced by sedge meadows in the elevated parts of the relief (*Carex aquatilis*, *Carex acuta*), whereas wet areas with sandy sediments are overgrown with horsetail (*Equisetum fluviatile*). Meadows of sedge and reed grass (*Calamagrostis langsdorffii*) are confined to the floodplain areas of the middle level, which are flooded for a shorter period, and meadows of reed grass, with mesophytic herbs, are formed at local micro-elevations. The thickets of willows (*Salix lapponum*, *S. lanata*, and *S. phylicifolia*) occupy meso-elevations. The areas of the ancient floodplain are characterized by an increase in the cover of moss in phytocoenoses [81].

## 4. Materials and Methods

### 4.1. Study Site

The study area is located in the northern part of the Pur-Taz interfluve (West Siberian plain, Russia), near the town of Tazovsky, in the continuous permafrost zone (67.35–67.43° N, 78.60–78.73° E) (Figure 12). The climate is humid semi-continental with mean annual temperature of –9.1 °C and annual precipitation of 360 mm. Abundant thermokarst lakes have developed on the third lake-alluvial terrace of the Taz River (30–40 m above the sea level). Sediments of the terrace consist of silty sandy loam, loam, and, less often, clay and sand. Lakes and khasyreys are distributed within the flat residual surfaces of the terrace. The khasyreys cover from 8.5% to 19.6% of the study area, whereas the lake coverage ranges from 4.8% to 7.6% [75]. Many of these khasyreys have been drained relatively recently, fewer than 100 years ago and constitute about 5% of the overall area [38]. The hydrochemical parameters of the lakes were studied by Serikova et al. [82].

The study area belongs to the WSL low-Arctic tundra. The ecosystems are represented by lichen-dwarf shrub tussock tundra on Cryosols (soil names to the World Reference Base for Soil Resources (WRB) [83]). Polygonal bogs and thermokarst lakes are confined to the wide depressions of the relief (Figure 13a). The bog soils are Dystric Cryic Histosols [84,85]. Thermokarst lakes are shallow, with a depth of 1 to 2 m. The bottom of the lake is flat, but there is usually a slope either towards the center of the basin or towards one of the edges. Since a large part of the studied area is waterlogged (up to 34%, according to Golubyatnikov et al. [75]), large lakes wash away the peat deposits of the adjacent bogs, which leads to an enrichment of the lake sediment with redeposited peat [86]. As a result, the share of peat in sediment increases towards the center of the lake. A shallow zone of coarse grain sand sediment forms along the mineral shore. Sometimes a sandbank is present near the shore, covering the peat sediments.

### 4.2. Dating the Drainage of the Studied Khasyreys

The drainage time was estimated using topographic survey materials from the late 1940s (Figure 14a), Landsat 4–8 satellite images of different years (Figure 14b), and complemented by radiocarbon dating of peat in khasyreys. The radiocarbon age of peat was determined by the liquid scintillation method using a Quantulus 1220 radiometer (Wallac, Turku, Finland) in the laboratory of the Tomsk Collective Use Center (Russian Academy of Sciences). Calibration of radiocarbon age was performed by the CALIB REV-7.10 program (Seattle, WA, USA).

Three studied early stage basins (Ea), namely Ea1, Ea2 and Ea3, started to drain quite actively over past 15 to 20 years. The drainage of Ea2 and Ea3 basins is still active and the majority of both basins are still covered by water. The permafrost has not yet formed in the basins, there is no cryogenic mound, and the thickness if organic layer is a few cm. The micro-relief has gentle slope to the center and coastal ripples. Figure 14b demonstrates gradual decrease of the lake water area from the Landsat 3–7 data.

The drained parts of the Mi1 and Mi2 basins (Mi—middle stage) are already present in the Landsat images taken 40 years ago, but on the topographical maps (completed at the end of 1940s), they were entirely covered by water. Therefore, their drainage must have occurred between 1950 and 1980, i.e., ≥50 y.a. These basins exhibit the appearance of cryogenic mounds, with maximal altitude difference of 1 m from the mound basement. More importantly, these basins already have moss-grass layers, those thickness exceeds 10 cm in wet depressions. Gramineous plants are not extensively developed and the role of mosses is higher than in the early stages.

Finally, the basins of the late (La) stage La1, La2 and La3 do not have lakes on the topographical map (Figure 14a). Field observations confirmed their attribution to the late stage due to well-developed *Sphagnum* cover and thick peat layer as well as low ALT. These basins are old (300 to 2000 y.o.), but not ancient (2000 to 5000 y.o.) because they do not have typical polygonal structures. To further validate these observations, we used ^14^C dating of the lowest part of peat cores, sampled in the middle of the basin, at the contact between peat and lake sediment. The beginning of peat formation in La1, La2 and La3 was 1083 ± 92 calBP, 303 ± 137 calBP, and 1761 ± 81 calBP.

Based on the time of lake drainage and field classifications [56,87], we divided all the studied khasyreys and their plant communities into three succession stages (Figure 13). These three stages are consistent with classes of basin ages as proposed by Hinkel et al. [87]. The Early stage corresponds to the first 50 years after drainage. The sites are characterized by the deepest ALT, typically between 1–1.5 m greater than that in the surrounding tundra ecosystems. The soils are represented by khasyrey sediments, with the first signs of aerobic development (oxidized layers, with an accumulation of plant litter). The Early stage corresponds to the Young class of Hinkel et al. [87]. The second (Mid) stage corresponds to a time of 50 to 200 years after drainage, or Medium class [87]. At this stage, organic soil horizons are accumulating, and the permafrost aggradation is well pronounced. The third (Late) stage corresponds to 200–2000 years after drainage, or the Old class of [87]. The ecotopes are characterized by an accumulation of the Fibric horizon on the soil surface. At this stage, the ALT in most parts of the basin is similar to that in the surrounding tundra ecosystems. The basins of the fourth, Ancient class (>2000 years old, [87], were not studied in this work).

### 4.3. Vegetation Properties

Using drone and satellite imagery, we identified 16 ecosystems in eight khasyreys representing the three different age classes. For a detailed characterization of the phytocoenotic diversity, each of 16 ecosystems was studied in two–four replicates (42 sites in total). At all study sites, we quantified the vegetation biomass and diversity and estimated the ecological conditions of the habitats, such as the soil moisture and degree of soil enrichment in nutrients. For this, we used indicator values of the plants, developed for the tundra and taiga vegetation zones of Western Siberia [88]. These values were also used to attribute the plant species to various ecological groups (listed in Appendix A: Figure A1; and Appendix B: Table A1).

The diversity of the ecosystems is primarily linked to different soil and topography conditions, which are, in turn, defined by the timing of the drainage. The characterization of vegetation included a list of the species, with their relative abundance, which was performed on 10 × 10 m plots or within natural contours (10–100 m^2^) in the case of small-sized plant communities. The productivity parameters of the plant communities were measured at several sites, representing each stage of the khasyrey succession. The aboveground biomass pool was determined only for ecosystems without shrubs. To estimate the site productivity, we harvested the above-ground biomass of the herb layer in 3 representative subplots of 0.25 m^2^ and recalculated the obtained values to 1 m^2^. The annual aboveground net primary productivity (ANPP, g m^−2^ year^−1^) was quantified as the sum of the dry weight of graminoids (Poaceae, Cyperaceae) and non-graminoid herbs (detailed description is provided in Axmanová et al. [89]). The measurement of the belowground biomass was performed by sequential collection of soil monoliths of 5 × 5 × 5 cm in 3 replicates from the surface, down to a depth of 30 cm, with a sampling step of 5 cm. For all aspects of the sample collection and handling, we followed the standard procedures, as described in Peregon et al. [90].

Field measurements of plant phytomass and productivity were compared with NDVI values which are used as a proxy for productivity [91]. Following common methods, we calculated NDVImax from all available Sentinel-2A imagery of the study area within the timeframe of when field work was conducted (Figure 13g,h). NDVImax represents the maximum NDVI values within a growing season [9,10,11,74,92]. The maximal values were selected based on all available images during the entire vegetation period of 2018 when the fieldwork was conducted. For this, we used a Feature Info Service chart software instrument. The cloud coverage was estimated using a relevant function in the EO browser and visually validated in the regime’s True color. We used the image was taken on 23 July 2018, which demonstrated the highest NDVI values, consistent with the time of our fieldwork (end of July to beginning of August). This corresponded to the highest field-based ANPP. The NDVImax were extracted from selected images using the Feature Info Service chart and Mark point. For this, we uploaded the GPS-based locations of each site in the .kml format. Thus obtained NDVI data corresponded to the pixel size of 10 m.

### 4.4. Biogeochemical Properties and Microtopography

Microtopographic evolution was studied by visual assessment in the field [93]. The altitude differences between mound and depression within the khasyreys were measured with total station Nikon Nivo 3.C (Nikon-Trimble, Tokyo, Japan). Soil profiles were studied using the shovel excavation method if the soil was not flooded. The soil flooded with water was sampled by a peat corer. Morphological description of the soil was performed in the field [93]. And the soil profiles were diagnosed by WRB [83]. Information on the diagnosed references soil groups is given in Table A2 and Figure A2. The ALT was measured using a probe 1.5 m long (Eijkelkamp, Giesbeek, The Netherlands). The soil density required to calculate the pool of elements was determined by a cutting ring method.

The trophic status of ecotopes was determined at 28 sites (typically 3–4 sites per khasyrey). The pH and specific electrical conductivity (Sp. cond.) of the soil water, sampled at the depth of the root layer, were measured in the field using Multi 3510 IDS portable instruments (WTW, Weilheim, Germany), following standard techniques [84,85]. The measurements were made in gravitational water-saturated soils, after digging a pit with a depth of 20 cm from the surface, which was filled by suprapermafrost water.

The potassium, nitrogen, and phosphorus (K, N, P) concentration and pool in soils were determined. For this, the soils were sampled in the rooting zone (0–30 cm) and separated into layers according to the soil horizons. The concentration of labile potassium, phosphorus, and nitrogen in soils was determined using standard methods [94]. The labile phosphorus fraction (P-PO_4_) was extracted from the soil samples using 0.2 M HCl. Extracted phosphorus was determined using a blue phosphorus-molybdenum complex on a UNICO 2100 spectrophotometer (Dayton, NJ, USA). In the same extract, potassium was also measured using an atomic absorption spectrometer equipped with the Kvant-2AT flame atomization device (Kortek, Moscow, Russia). Ammonium (N-NH_4_) and nitrate (N-NO_3_) were extracted from the samples using Milli-Q deionized water (30 g of soil per 75 mL of MilliQ water during 24 h), then determined on the UNICO 2100 spectrophotometer. The N-NH_4_ was measured using mercuric ammonium iodide, and N-NO_3_ was analyzed with phenol disulfonic acid. The nutrient pools (g·m^−2^) were calculated for a 0–30 cm layer using soil density data, determined by the cutting ring method in each of the horizons.

The elementary composition of the plant biomass was determined by averaging three samples of the aerial parts of the plants, after dry biomass grinding using the planetary mill (Retsch GmbH, Haan, Germany). The total elementary composition of plant biomass was determined after full acid digestion, using the ICP-MS analyses (7500 ce, Agilent, Santa Clara, CA, USA) was employed (see details in Stepanova et al. [95] and Viers et al. [96]). The content of C and N in the biomass was determined using a Thermo Flash 2000 analyzer (Thermo Fisher Scientific, Waltham, MA, USA), calibrated using aspartic acid.

### 4.5. Statistical Analyses

Statistical treatment of a complete set of the productivity and soil and plant chemical composition included principal component analysis (PCA) using the Statistica-12 package (StatSoft, Tulsa, OK, USA), containing methods for scores and variables [97]. The data exhibited an non-normal distribution as inferred from the Kolmogorov-Smirnov and Lilliefors test for normality and the Shapiro-Wilk’s test. Non-parametric H-criterion Kruskal Wallis and Mann-Whitney U-test were used to validate the differences between succession stages of vegetation. The PCA made it possible to characterize the dependence of the aboveground net primary production and NDVImax on various environmental variables, such as the peat thickness, active layer thickness, Sp. cond. and pH of the soil water, soil density, labile phosphorus, mineral nitrogen and potassium concentration in the soil. The plots were created using MS Excel 2016, MS Visio Professional 2016 (Microsoft, Redmond, WA, USA), the GS Grapher 13 package (Golden Software, Golden, CO, USA) and the CorelDRAW Technical Suite 2017 (Corel Corporation, Ottawa, ON, Canada).

## 5. Conclusions

The vegetation of the khasyreys of the Pur-Taz interfluve is controlled by properties of the lake sediment and bottom microtopography. This is expressed in the species composition, ecological structure and productivity of plant communities. After the initial drainage of the lake, the main areas of the basin exhibit wet mesotrophic conditions. These ecotopes occupy sedge meadows with a dominance of *Carex rostrata* or *C. aquatilis* and the participation of other hydrophilic herbs demanding soil trophicity. In the wettest and the most nutrient-rich ecotopes, the phytocoenoses with a dominance of *Arctophila fulva* are formed. *Equisetum fluviatile* forms thickets on the nutrient-poor, over-wet ecotopes. In more drained areas, the abundance of *Calamagrostis langsdorfii* increases. On the top of frost mounds, this reed grass forms meadows with *Equisetum arvense* and the participation of mesophytic herbs. The vegetation of khasyreys is further differentiated in the course of time. During the early stage of the colonization of lake basins (first decades), phytocoenoses with a well-developed herb layer are formed. At the mid successional stage (≥50 years after drainage), herbaceous plant communities are formed, in which a moss layer is also developed. Initially, the moss layer consists of hydrophilic brown mosses (Bryales); with time, sphagnum mosses enter the plant communities. At the late successional stage of khasyreys (several hundred years after the lake drainage), a thick peat layer is accumulated, and wet mesooligotrophic conditions develop. These ecotopes are occupied by communities of sphagnum fens, consisting of species that demand less soil trophicity. *Carex limosa*, *C. rotundata*, *C. chordorrhiza* and *Eriophorum russeolum* compose a sparse herb layer. *Sphagnum balticum*, *S. jensenii* or *S. majus* dominate in the standing fens, and *Sphagnum obtusum* dominates in the flowing fens. Dwarf shrub-moss-lichen communities form on the frost mounds.

Overall, the formation of highly productive herbaceous communities at the early stage of the lake bottom is largely controlled by the availability of soil nutrients to plants, which is primarily determined by the thickness of the active layer and the accumulation of organogenic horizons. Due to the nutrient-rich soils of the early and mid khasyrey, the plant communities at these stages are characterized by a high productivity, which is several times higher than the productivity of the background tundra communities. With thickening of the organogenic horizons and the simultaneous depletion of some soil nutrients, a decrease in the phytocoenoses productivity is observed, which is already noticeable at the mid successional stage. Plant communities of the late successional stage are characterized by a minimal productivity, since a thick peat layer prevents the plants from absorbing nutrients from the lake sediment. The grass litter accumulation and its transformation into peat are the main processes controlling the vegetation succession of khasyreys. The lake basins at the late successional stage are characterized by a very low productivity, and they are similar to the polygonal bogs in terms of species composition and ecological structure. Young and mid-stage khasyrey are capable to greatly contributing to the observed greening of the northern WSL and thus require further extensive studies across the other regions of the Arctic tundra.

## Figures and Tables

**Figure 1 plants-09-00867-f001:**
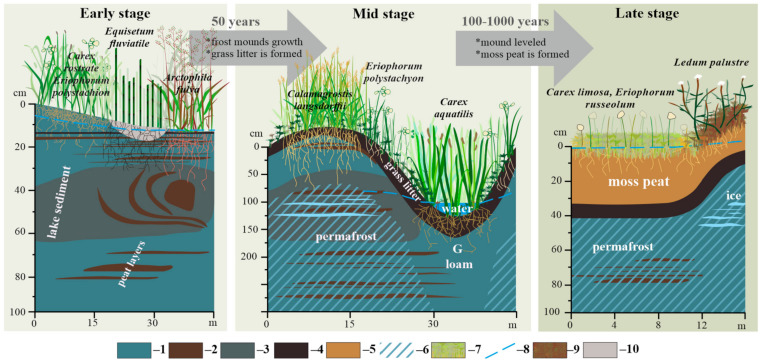
Vegetation and soil of the successional stages of khasyreys. 1—mineral lake sediment; 2—inclusion of redeposited peat in the sediment; 3—peat–mineral sediment; 4—grass (eutric) peat; 5—moss (dystric) peat; 6—permafrost; 7—sphagnum mosses; 8—top level of shallow ground water; 9—brown and sphagnum moss; 10—sandy layer of sediment.

**Figure 2 plants-09-00867-f002:**
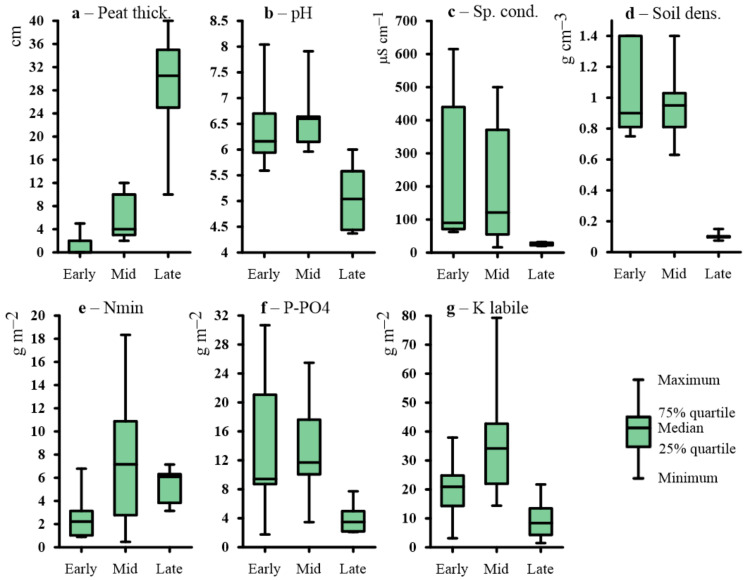
Analytical properties in a soil layer of 0–30 cm through three successive stages of khasyreys. (**a**)—peat thickness; (**b**)—soil water pH; (**c**)—specific electrical conductivity; (**d**)—soil density; (**e**)—mineral forms of nitrogen (N-NO_3_^−^ + N-NH_4_^+^); (**f**)—labile phosphorus pool in 0–30 cm layer; (**g**)—labile potassium pool in the 0–30 cm layer.

**Figure 3 plants-09-00867-f003:**
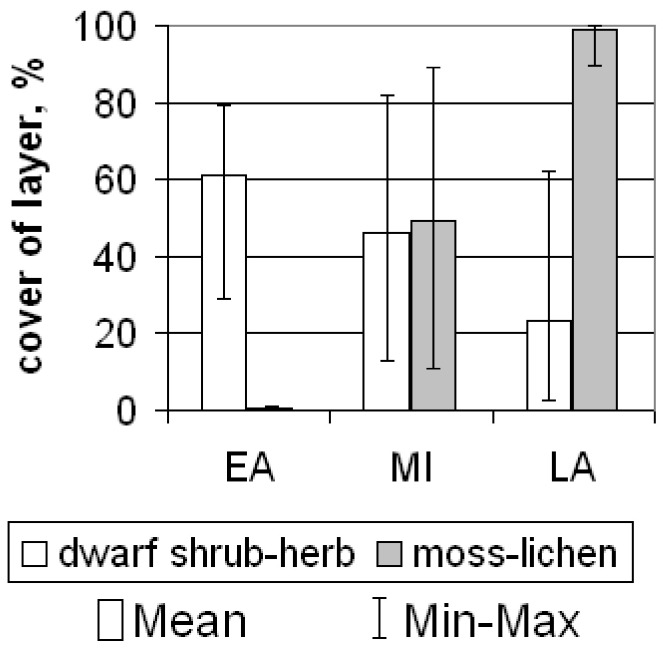
The plant communities of the Early (EA), Mid (MI) and Late (LA) successional stages.

**Figure 4 plants-09-00867-f004:**
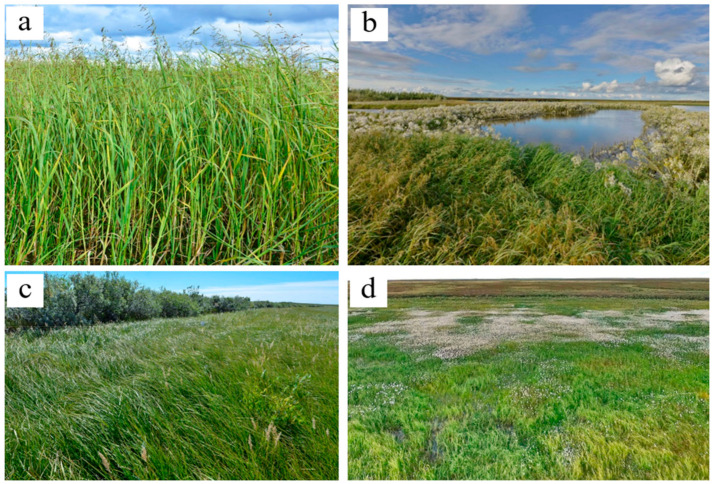
The vegetation of the early khasyreys: (**a**)—*Arctophila fulva*; (**b**)—*Arctophila fulva* and *Tephroseris palustris* around the pond; (**c**)—community on the sandbanks near the shore; (**d**)—complex vegetation cover of the early khasyrey (green—sedge; white—cotton grass).

**Figure 5 plants-09-00867-f005:**
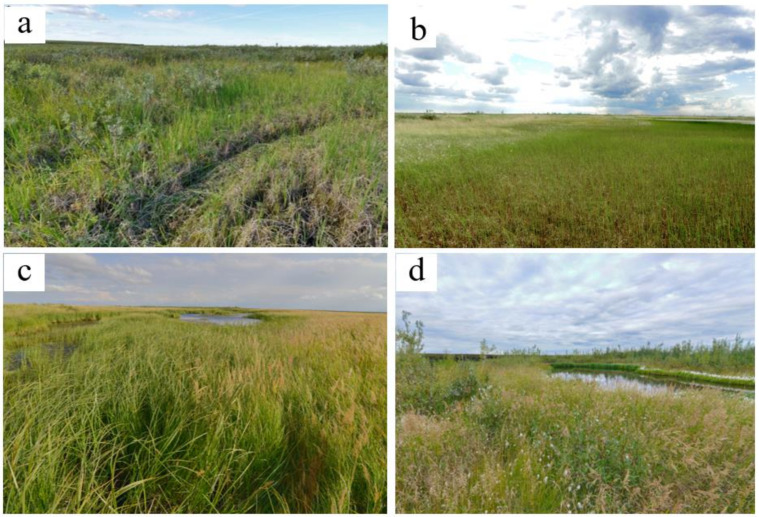
The vegetation of the mid khasyreys: (**a**)—complex vegetation of frost mounds (birch-brown moss community) and hollows (sedge brown moss community); (**b**)—horsetail-brown moss community (**c**)—sedge and reedgrass community on a slope to permafrost subsidence; (**d**)—reed-willow community and residual pond.

**Figure 6 plants-09-00867-f006:**
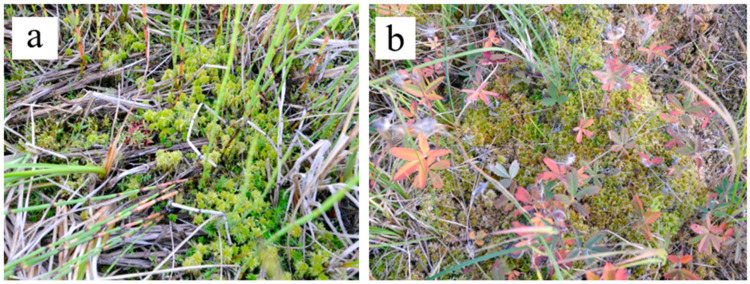
Patches of sphagnum mosses in horsetail-sedge (**a**) and sedge-comarum communities (**b**).

**Figure 7 plants-09-00867-f007:**
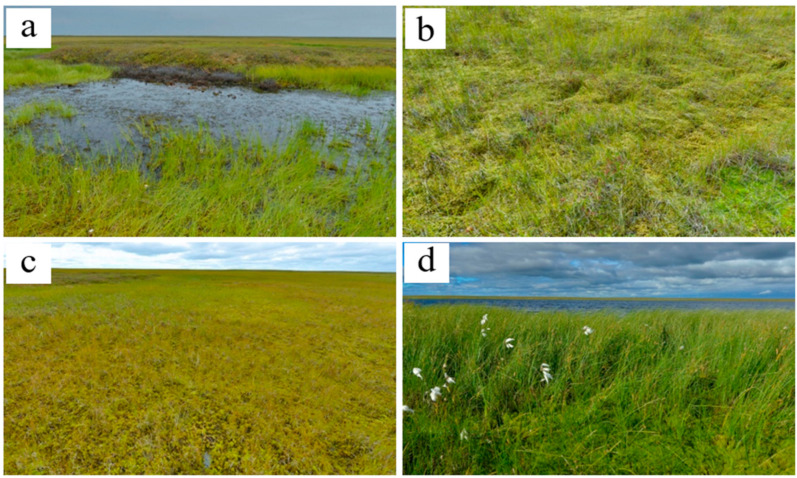
The vegetation of the late khasyreys: (**a**)—frost mound with a dominance of ledum and lichen, and the thawed pond with *Eriophorum* spp.; (**b**)—standing fen with *Carex limosa*, *Carex chordorrhiza* and *Sphagnum obtusum*; (**c**)—flowing fen with *Carex chordorrhiza*, *Sphagnum obtusum*, *Calliergon stramineum* and *Warnstorfia exannulata*; (**d**)—shore of residual pond with *Carex rostrasta* and *Eriophorum medium*.

**Figure 8 plants-09-00867-f008:**
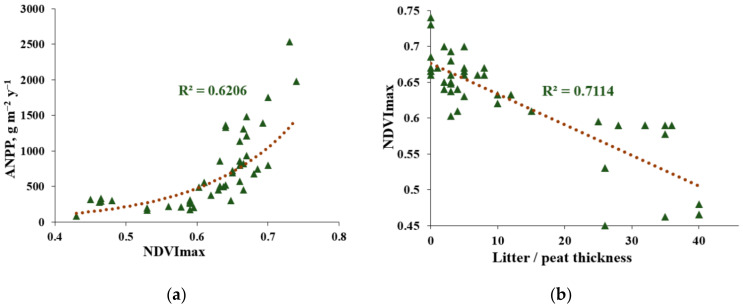
Relationship between the NDVImax and the ANPP (**a**) and litter: peat thickness ratio (**b**) of all sites studied. The NDVImax measured using EO Browser for the year of field work.

**Figure 9 plants-09-00867-f009:**
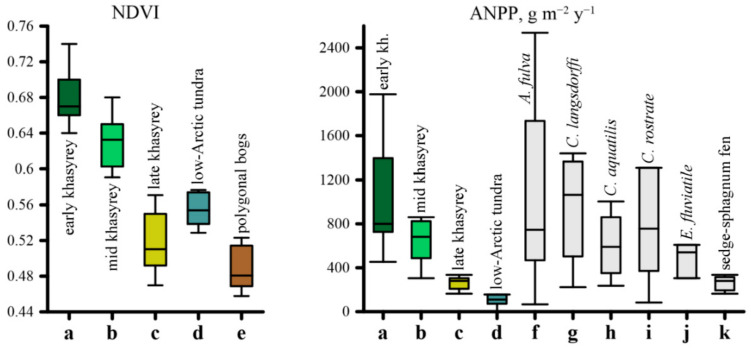
The variability of the aboveground net primary productivity and NDVImax (median, percentiles, minima and maxima) in khasyrey plant communities: a, b, c—early, mid and late stage successions of plant communities, respectively; d—plant communities of low-Arctic tundra (ANPP are taken from Bazilevich [67,68]); e—polygonal bogs; f, g, h, i, j—plant communities dominated by *Arctophila fulva*, *Calamagrostis langsdorffi*, *Carex aquatilis*, *Carex rostrate*, *Equisetum fluviatile*; k—sedge-cotton grass-sphagnum fen.

**Figure 10 plants-09-00867-f010:**
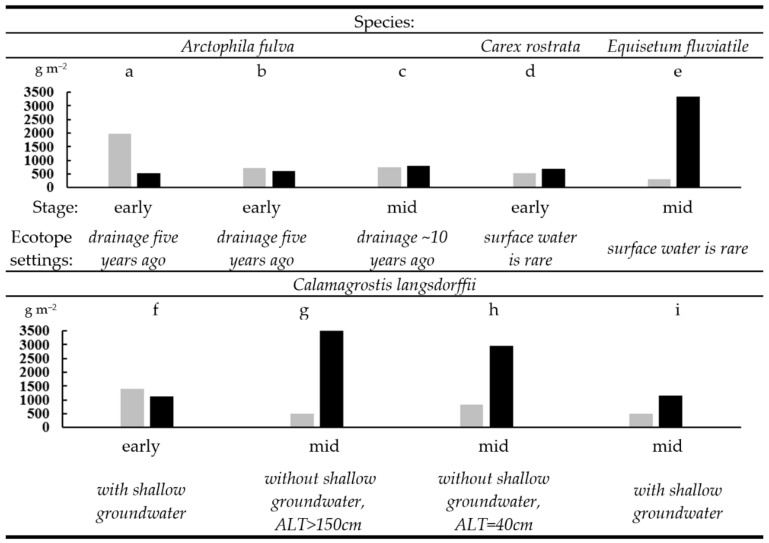
The ratio of aboveground (grey) and underground (black) phytomass (g·m^−2^). The meadows dominated by: (**a**–**c**)—*Arctophila fulva* (sites tz18–28, tz18–47, and tz18–24, respectively); (**d**)—*Carex rostrata* (tz18–46); (**e**)—*Equisetum fluviatile* (tz18–51); (**f**–**i**)—*Calamagrostis langsdorffii* (tz18–27, tz18–50, tz18–52, and tz18–53, respectively).

**Figure 11 plants-09-00867-f011:**
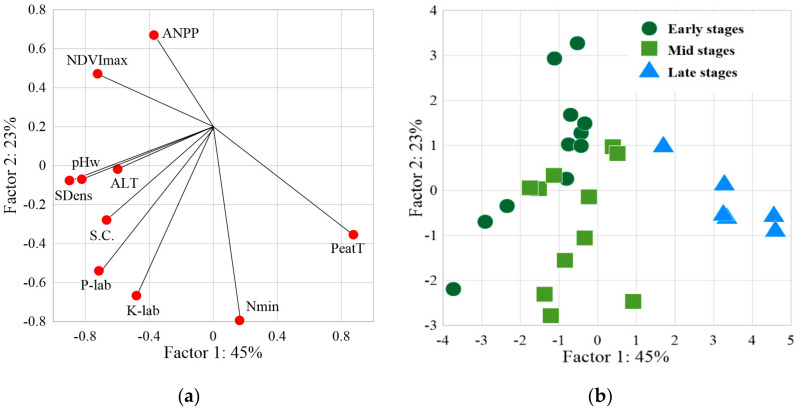
Results of the PCA treatment of the soil trophic parameters and vegetation productivity: (**a**)—the relationship between the input parameters and 2 possible factors (ANPP—aboveground net primary production; NDVImax—average maximum NDVI value for 2016–2019; ALT active layer thickness; pHw—pH of soil water; S.C.—specific electrical conductivity; SDens—soil density; P-lab—labile phosphorus; K-lab potassium; Nmin—mineral compounds of nitrogen; PeatT—peat thickness), (**b**)—studied sites of three successional stages of the khasyreys in relation to possible factors (stages: Ea—early; Mi—mid; La—late).

**Figure 12 plants-09-00867-f012:**
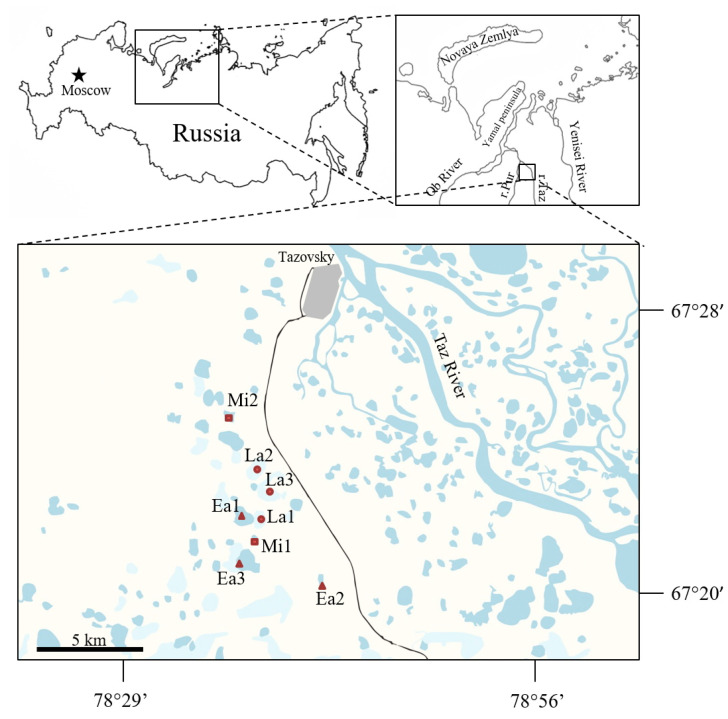
Map of the research area in the northern part of the Pur-Taz interfluve (northern WSL). The studied khasyreys different stages are shown (Ea—Early, Mi—Mid, La—Late).

**Figure 13 plants-09-00867-f013:**
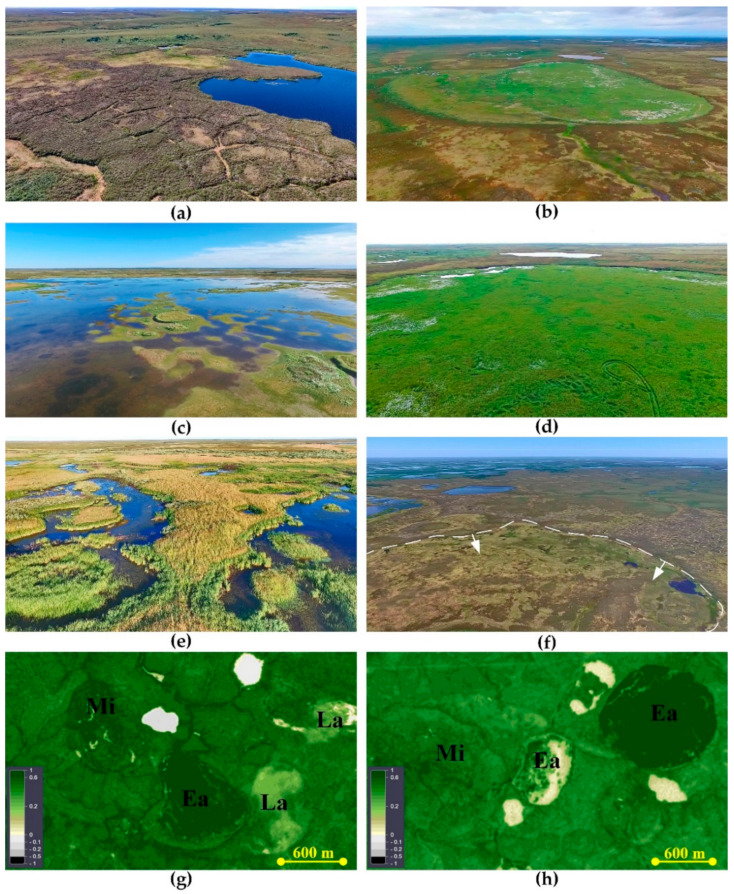
Landscape and NDVI values in northern part of the Pur-Taz interfluve. (**a**) Tundra and polygonal frozen bog; (**b, c, d**) the early khasyrey; (**e**) mid khasyrey; (**f**) late khasyrey; (**g**) and (**h**) NDVI values from modified Copernicus Sentinel data ((**g**)—08.22.2018 and (**h**)—20.07.2019), processed by the EO Browser (abbreviations for succession stages see Figure 12).

**Figure 14 plants-09-00867-f014:**
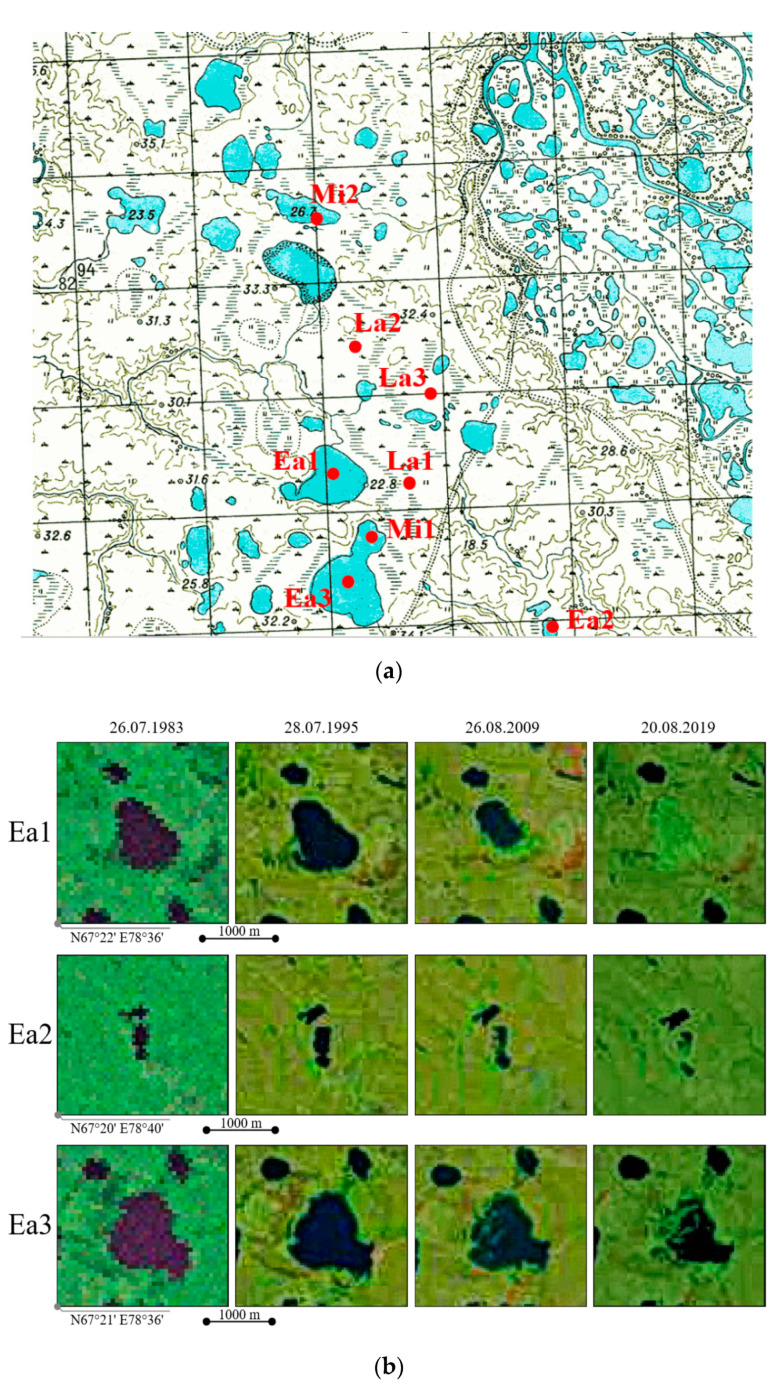
Map of the Tazovsky key site of the late 40 s (**a**) and satellite images of Landsat 3–7 with early khasyrey (**b**).

**Table 1 plants-09-00867-t001:** Mean values and standard deviations (M ± sd) of soil and vegetation parameters.

Successional Stages	Early	Mid	Late
NDVImax	0.68 ± 0.03	0.64 ± 0.02	0.54 ± 0.07
ANPP, g·m^−2^·y^−1^	1134 ± 645	660 ± 292	261 ± 77
Litter/peat thickness, cm	1.3 ± 1.7	6.2 ± 4.3	28.5 ± 10.7
ALT, m	1.53 ± 0.58	1.14 ± 0.71	0.69 ± 0.65
Soil density, g·cm^−3^ (0–30 cm)	1.0 ± 0.3	1.0 ± 0.3	0.1 ± 0.02
pH soil water	6.3 ± 0.7	6.5 ± 0.7	5.1 ± 0.7
Sp. cond. of soil water, µS·cm^−1^	208 ± 205	171 ± 170	25 ± 5
P-PO4, g·m^−2^ (0–30 cm)	13.8 ± 8.4	13.3 ± 6.2	4.0 ± 2.2
K labile, g·m^−2^ (0–30 cm)	20.8 ± 10.1	35.8 ± 18.2	9.6 ± 7.2
N min (0–30 cm)	2.5 ± 1.8	7.4 ± 5.5	5.4 ± 1.6
Cover of herbs, %	61.1 ± 18.0	46.2 ± 22.4	5.0 ± 5.0
Cover of mosses, %	0.7 ± 0.6	50.8 ± 26.0	99.0 ± 2.3

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
