# Peer review of "Lake Drainage in Permafrost Regions Produces Variable Plant Communities of High Biomass and Productivity"

_plants, 2020, doi:10.3390/plants9070867_

Round 1
Reviewer 1 Report
Manuscript was strengthened by demonstrating statistical differences among basin ages. Minor word and sentence edits throughout the manuscript are needed. The intro is informative and will benefit from some sentence editing to improve flow. New figures need some minor edits for reader clarity and figure descriptions can be improved. Below, I added some suggestions but additional edits/improvements can be done by the Authors discretion to polish the manuscript and figures.
Other minor edits:
181- missing author. Please add last name of author (not just reference): “..as proposed by Hinkel et al. [70].”
191- missing closing parenthesis
240- please clarify what does Z hrs mean
390- spell check for “wre” and modify sentence: “between different stages” to improve flow.
Figure 11- I will suggest to add a legend to figure to indicate the different plots besides just mentioning a,b,c…etc in the footnote (which can be confusing for the readers). Or perhaps and abbreviation of the plot name such as early, mid, late, etc. or use vertical labels on the x axis.
Fig 12. The figure can be modify into a vertical bar graph that includes biomass in the Y axes and sites in the x axes and a consistent scale across all on the Y axes. I will recommend to label it as site name vs “a, b, c,” and keeping the dominant plant species and ecotope settings. Also, I am a bit confused about the term “top water”, is this surface water or inundation? please clarify on figure and be consistent throughout.
Fig 13- I will recommend to use different point shapes & color for each basin age for clarity (i.e. triangles/blue, squares/green, circles/red). Names next to points seem a bit difficult for identifying patterns in the plot.
Author Response
Reviewer 1
Manuscript was strengthened by demonstrating statistical differences among basin ages. Minor word and sentence edits throughout the manuscript are needed. The intro is informative and will benefit from some sentence editing to improve flow. New figures need some minor edits for reader clarity and figure descriptions can be improved. Below, I added some suggestions but additional edits/improvements can be done by the Authors discretion to polish the manuscript and figures. – – We are grateful for positive evaluation of our work and we edited and polished the text, figures and their captions and legends.
Other minor edits:
181- missing author. Please add last name of author (not just reference): “..as proposed by Hinkel et al. [70].”
– Corrected
191- missing closing parenthesis
– Corrected
240- please clarify what does Z hrs mean
– Clarified
390- spell check for “wre” and modify sentence: “between different stages” to improve flow. – – – Corrected
Figure 11- I will suggest to add a legend to figure to indicate the different plots besides just mentioning a,b,c…etc in the footnote (which can be confusing for the readers). Or perhaps and abbreviation of the plot name such as early, mid, late, etc. or use vertical labels on the x axis.
– We followed this valuable advice and revised abbreviations and added labelling to the plot for better clarity
Fig 12. The figure can be modify into a vertical bar graph that includes biomass in the Y axes and sites in the x axes and a consistent scale across all on the Y axes. I will recommend to label it as site name vs “a, b, c,” and keeping the dominant plant species and ecotope settings. Also, I am a bit confused about the term “top water”, is this surface water or inundation? please clarify on figure and be consistent throughout.
– We greatly reorganized this figure and presented it as vertical bar graph, as recommended. We replaced the term “top water” by “shallow groundwater”
Fig 13- I will recommend to use different point shapes & color for each basin age for clarity (i.e. triangles/blue, squares/green, circles/red). Names next to points seem a bit difficult for identifying patterns in the plot.
– We thank the reviewer for these valuable suggestions and greatly revised this figure as recommended.
We are grateful to reviewer No 1 for very constructive comments

Reviewer 2 Report
This study entitled “Lake drainage in permafrost regions produces variable plant communities of a high biomass and biological productivity” examines vegetation succession of drained thermokarst lake basins in the WSL using a space-for-time continuum. The authors provide an extremely detailed characterization of species composition, biomass and biochemical soil properties of three categories of drained lake basins or khasyreys. NDVI is also extracted from Sentinel-2. A PCA was used to examine the relationships between vegetation variables and environmental variables. The study is timely and an important contribution to the body of research trying to provide a more detailed understanding of the ecosystem processes driving large-scale NDVI trends in the Arctic. While the authors provide a comprehensive dataset and analyses, the lack of clarity and poor organization of the paper takes away the impact it could and should have. There are grammar issues that need resolving. I have provided detailed corrections of language mistakes for the Abstract and Introduction as well as suggestions for restructuring to improve clarity. I highly suggest a native English speaker proof-read and edit any future versions of the paper. While the potential for a really excellent paper is there, given the scope of restructuring I believe is necessary, I suggest a major revision of the paper before it is considered for publication in Plants. Please see my extensive and detailed comments below.
Title:
I would remove biological as you are only referring to primary productivity and productivity implies vegetation productivity.
Abstract
The abstract needs to be better organized. In general, abstracts begin with a motivation, then a brief introduction, summary of key methods and then the most important results. This is then followed by a brief summary of what the results suggest with a few key discussion points. Finally, a concluding sentence highlighting the impact or situating the results in the broader research context (such a sentence is currently missing). Right now, these sections are not well separated and often mixed making it difficult to easily understand what you did and how. The methods are too brief, how do you actually measure productivity? The results need to be more quantitative and concise. Save the explanation of the results (e.g., contributing to greening) for later as a key discussion point. Remember the abstract is the first, and often the most widely read part of your paper so it is imperative that it is clear and concise.
Introduction
In general, I feel the Intro could benefit from a restructuring. For example, in the first paragraph you introduce the topic of NDVI (without mentioning remote sensing or satellites – this comes only at the end of the paragraph) in the second sentence and then it comes up again near the end of the paragraph after you talk about the specific trends of the WSL.
I would try to restructure this paragraph in the following way and the remaining paragraphs should follow a similar restructuring:
Introduce the problem: longer, warmer growing seasons, more available nutrients and greater precipitation leading to increased productivity and increased thermokarst activity. This paper focuses on thermokarst lake drainage. Introduce this right away.
What we know: the effect of increased productivity is known as tundra greening and has been confirmed both by satellite-derived NDVI and in-situ research (citations). This is not only caused by increased vegetation activity, but more importantly by colonization of previously unvegetated drained lake basins.
Specifics about the area of interest: here introduce the WSL greening stats and lake basin drainage info.
The way you describe your region of interest, the northern part of the WSL is inconsistent making it difficult to understand some of the information you present in this section. My suggestion is to define right away what part of the WSL you are talking about and then name it the northern WSL, and not use “territory” or “northern part of the WSL”. This will clarify your message. This is further confused by the use of “southern part of the WSL tundra zone” in the methods. Please be sure to clarify and be consistent throughout the manuscript.
Materials and Methods
I would like to see Figure A1 in the body of the text as it is referred to multiple times and is a key illustration of your dating methods.
Section 2.3 definitely needs to be split into two sections as they are quite distinct (1. Vegetation properties and 2. biochemical properties and microtopography). This will also clarify your methods which are currently presented in a disjointed and confusing way.
A schematic of your sampling design would be helpful.
How exactly is NDVI max calculated? How many pixels, how did you deal with data gaps or cloud cover? What level of product did you use? Was there a threshold for the timeframe of the data? What I mean is when peak season images weren’t available did you then use scenes from within a specific date range? When not, how did you account for that?
Results
Table 1 should be moved to the appendix. It is just far too much information for one table in the body of the text.
Table 3 should be moved to the appendix and or only a few key nutrients (e.g., C, N, P) should be presented, not all of them.
Discussion
There is a lot of information presented in this study and I fear that the key results and discussion points are a bit lost in the incredible amount of information presented in the discussion. With this in mind, I would suggest rewriting the discussion with fewer specifics and more general trends. I really like the way section 4.1 begins with the 4 key factors influencing vegetation succession. However, you do not need to reiterate how the Khasyreys form (Line 500-506). At this point we have heard this information in great detail more than once. Additionally, rather than reiterating results (e.g., Line 508) you can simply say that soil fertility is higher in early and mid-stages which favors the formation of specific plant communities which produce high phytomass. When you talk about flood resistance, we don’t need specific species ratios, only that greater duration of flooding drives succession due to shifting Carex abundance. With this change in mind, I would like to see each paragraph rewritten in a similar way to distill the key points while leaving out previously reported results. I also suggest that the final section be significantly shortened. It is good information but can be presented in a much more concise manner so we as the reader can understand how representative the system is at the Arctic biome scale.
Specific comments
Line 19: change to “frequency of full or partial drainage…”
Line 20: remove abundant, not the right word here, maybe productive or highly vegetated
Line 21: remove extremely
Line 21: poorly understood
Line 21-24: combine these sentences “We characterized vegetation communities and productivity of drained thermoskarst lake basins (khasyreys) in the continuous permafrost zone of the Western Siberian Lowland (WSL), the largest peatland in the world.”
Line 25: Productivity of khasyreys vegetation was two to nine times higher than that of surrounding tundra.
Line 26-27: I would move this greening part to later in the abstract – it doesn’t fit here because there is no previous mention of this idea/theory. This is a discussion point so I would put it that part of the abstract
Line 27: No previous mention of successional stage. If this is part of your methodology, please mention it in the methods.
Line 28: Also no previous mention of the active layer being monitored or considered in the study – please also mention this briefly in the outline of the methods.
Line 31: change to “and nutrients are leached…”
Line 36: remove “on the soil surface”
Line 37: change to: Vascular plants, which cover between 10% and 60% of the
Area, are dominated by dwarf shrubs and the abundance of herbaceous species is minimal. Mosses and lichens have continuous coverage.
Line 38: tell us again which stage it is (late successional)
Line 41: remove “soil surface and microtopography of the lake bottom” this adds confusion, just having accumulation of peat is enough.
Line 50: change to “an increase in soil nutrient concentrations and precipitation”
Line 51: change to “… an increase in primary productivity…”
Line 52: I would say. “Increases in tundra productivity contributes to an effect known as tundra greening…” not all greening is an increase in productivity. You also need citations here and for the browning sentence (e.g. Frost et al., (2019) Tundra Greenness [In Arctic Report Card 2019]).
Line 55: what effects? Be specific and mention again greening and browning. Change “such as” to “including”
Line 56: change “it” to “trends”
Line 57: this wording is confusing, 18% of the entire territory of the northern part of the WSL. What are you talking about here? Does territory refer to a specific province of a specific size? The 18% is of the entire WSL or just a sub area? Please clarify
Line 58: what does pronounced mean? Also try to be consistent in your nomenclature, is territory really necessary? Can you not just use WSL?
Line: 59: remove “It was established that”
Line 62: change to “observations”
Line 62: This sentence is redundant and doesn’t really tell the reader anything.
Line 61-65: I think this should come before the specific information about the WSL.
Line 61-62: You already said this in Line 53
Line 67-68: seasonally unfrozen layer of the permafrost
Line 68-69: be more specific about what changes you are referring to – climatic and ground thermal regime changes. Also the wording here needs to change, permafrost stability does not respond by increasing thermokarsting. The changes in climatic and thermal regimes lead to increased thermokarsting.
Line 71: This description of how thermkarst lakes drain needs to be improved. Define trough, water channel and provide a more mechanistic description of the process. This is not sufficient
Line 73-74: I would add in here that sediments are relatively warm and nutrient rich. Also change forming, to “beginning primary succession…”
Line 74-79: move to Line 72. Talk about the trends of this phenomenon and then move on to the vegetation succession.
Line 77-79: You should mention how increasing lake size is connected to browning trends.
Line 81: change border to interface
Line 81: Thermokarst activity
Line 82: change growing, to deepening (change this language throughout the paragraph)
Line 86-87: I think you should either remove this sentence about the upland and polygonal bogs or add more mechanistic detail on how this happens.
Line 91: change to “major factors of tundra greening”
Line 93: change to “Nutrient-rich sediments favour…”. Remove “very”
Line 94: remove “Specifically” and change to “The drainage of thermokarst lakes…”
Line 96: observed by whom? Use occurs
Line 97: permafrost aggregation occurs differently in different parts of the basin
Line 97: what is permafrost development intensity? Depth? Temperature? Be specific
Line 98: further explanation and a citation is needed for this sentence.
Line 98: what does “The latter” refer to?
Line 103: How? Describe how vegetation influences spectral reflectance
Line 106: It would also be good to mention again why validation of NDVI trends is so important.
Line 117: remote “the”
Line 137: further confusion about the study area as mentioned above
Line 153: do you do the radiocarbon dating or is this from other work? If from others, please cite that work here.
Line 154: define Ea1, etc.
Line 160: define Mi1, etc.
Line 181: add author names to citation [70]
Line 192: split this section into two: Vegetation properties and Biochemcial properties and microtopography
Line 216: remove “each”
Line 218: change to (detailed description is provided in…) add authors name not ref.
Line227: remove “one”
Line 229: define WTW
Line 253: change to non-normal
Line 258: parameters refer to model outputs. Variables is the proper word to use here.
Line 276: Table 1. Lakes don’t have coasts, they have shores, please change this accordingly
Line 291: change to “throughout the entire basin”
Line 292: change to “on average”
Line 294: change encountered to present
Line 336: change to “which are common”
Line 339: change to “… like in the early…”
Table 3 can also be moved to appendix
Line 508: remove “U-Test”
Author Response
Reviewer 2
We are very grateful for insightful and constructive comments on our revised manuscript. We greatly revised the text, figures and tables following your suggestions.
I would remove biological as you are only referring to primary productivity and productivity implies vegetation productivity. - Corrected
Abstract
The abstract needs to be better organized. In general, abstracts begin with a motivation, then a brief introduction, summary of key methods and then the most important results. This is then followed by a brief summary of what the results suggest with a few key discussion points. Finally, a concluding sentence highlighting the impact or situating the results in the broader research context (such a sentence is currently missing). Right now, these sections are not well separated and often mixed making it difficult to easily understand what you did and how. The methods are too brief, how do you actually measure productivity? The results need to be more quantitative and concise. Save the explanation of the results (e.g., contributing to greening) for later as a key discussion point. Remember the abstract is the first, and often the most widely read part of your paper so it is imperative that it is clear and concise.
– We strongly shortened and focused the revised Abstract following the recommendations of reviewer. We would like to note that there is no word limit on the Abstract in this journal so we feel it affordable to have rather extended presentation of our results.
Introduction
In general, I feel the Intro could benefit from a restructuring. For example, in the first paragraph you introduce the topic of NDVI (without mentioning remote sensing or satellites – this comes only at the end of the paragraph) in the second sentence and then it comes up again near the end of the paragraph after you talk about the specific trends of the WSL. I would try to restructure this paragraph in the following way and the remaining paragraphs should follow a similar restructuring: Introduce the problem: longer, warmer growing seasons, more available nutrients and greater precipitation leading to increased productivity and increased thermokarst activity. This paper focuses on thermokarst lake drainage. Introduce this right away.
What we know: the effect of increased productivity is known as tundra greening and has been confirmed both by satellite-derived NDVI and in-situ research (citations). This is not only caused by increased vegetation activity, but more importantly by colonization of previously unvegetated drained lake basins.
Specifics about the area of interest: here introduce the WSL greening stats and lake basin drainage info.
– We reorganized the beginning of Introduction as recommended and added the sentences suggested by reviewer. We further removed non-essential information.
The way you describe your region of interest, the northern part of the WSL is inconsistent making it difficult to understand some of the information you present in this section. My suggestion is to define right away what part of the WSL you are talking about and then name it the northern WSL, and not use “territory” or “northern part of the WSL”. This will clarify your message. This is further confused by the use of “southern part of the WSL tundra zone” in the methods. Please be sure to clarify and be consistent throughout the manuscript.
– We agree that the territory can be defined as northern WSL and revised the text accordingly. However, we would like to keep the term “low-Arctic tundra” in places where it is needed for better understanding of ecosystem and biome context.
Materials and Methods
I would like to see Figure A1 in the body of the text as it is referred to multiple times and is a key illustration of your dating methods.
– We moved Figure A1 to the Materials and Methods section.
Section 2.3 definitely needs to be split into two sections as they are quite distinct (1. Vegetation properties and 2. biochemical properties and microtopography). This will also clarify your methods which are currently presented in a disjointed and confusing way.
– Thank you for your comment. We fully agree and divided this section into two distinct parts.
A schematic of your sampling design would be helpful.
– We presented the sampling scheme in the Graphical Abstract
How exactly is NDVI max calculated? How many pixels, how did you deal with data gaps or cloud cover? What level of product did you use? Was there a threshold for the timeframe of the data? What I mean is when peak season images weren’t available did you then use scenes from within a specific date range? When not, how did you account for that? How exactly is NDVI max calculated?
– We clarified the method of obtaining data on NDVImax and we Added the following text:
Field measurements of plant phytomass and productivity were compared with NDVI values which are used to assess gross photosynthesis [77]. Following the common practices [9-11,78,79] we used annual maximal NDVImax. These values characterize the maximal plant development during the vegetation period. The NDVI index data for each study site were obtained using Sentinel-2A satellite images, available in the Sentinel Hub EO Browser (Figure 2g,h). The maximal values were selected based on all available images during the entire vegetation period of 2018 when the fieldwork was conducted. For this, we used a Feature Info Service chart software instrument. The cloud coverage was estimated using a relevant function in the EO browser and visually validated in the regime's True color. We used the image was taken on July 23, 2018, which demonstrated the highest NDVI values, consistent with the time of our fieldwork (end of July -beginning of August). This corresponded to the highest field-based ANPP. The NDVImax were extracted from selected images using the Feature Info Service chart and Mark point. For this, we uploaded the GPS-based locations of each site in the .kml format. Thus obtained NDVI data corresponded to the pixel size of 10 m.
Results
Table 1 should be moved to the appendix. It is just far too much information for one table in the body of the text.
– We agree and corrected
Table 3 should be moved to the appendix and or only a few key nutrients (e.g., C, N, P) should be presented, not all of them.
– The table has been moved to the appendix.
Discussion
There is a lot of information presented in this study and I fear that the key results and discussion points are a bit lost in the incredible amount of information presented in the discussion. With this in mind, I would suggest rewriting the discussion with fewer specifics and more general trends. I really like the way section 4.1 begins with the 4 key factors influencing vegetation succession. However, you do not need to reiterate how the Khasyreys form (Line 500-506). At this point we have heard this information in great detail more than once. Additionally, rather than reiterating results (e.g., Line 508) you can simply say that soil fertility is higher in early and mid-stages which favors the formation of specific plant communities which produce high phytomass. When you talk about flood resistance, we don’t need specific species ratios, only that greater duration of flooding drives succession due to shifting Carex abundance. With this change in mind, I would like to see each paragraph rewritten in a similar way to distill the key points while leaving out previously reported results. I also suggest that the final section be significantly shortened. It is good information but can be presented in a much more concise manner so we as the reader can understand how representative the system is at the Arctic biome scale.
– We thank you for such significant and important comments that allowed us to greatly revise this section. We tried to reduce the discussion, based on the proposed scheme. Therefore we greatly revised section 4.1. When presenting our discussion, it is however essential to remind what kind of results are being discussed. Otherwise, in order to avoid results repetition, the paper should be restructured into one combined results and discussion section which is clearly not desirable at this stage. Note that sections 4.2 and 4.3 are less affected by re-presentation of results, although we revised the text in these sections as well.
Specific comments
Line 19: change to “frequency of full or partial drainage…”
– Fixed
Line 20: remove abundant, not the right word here, maybe productive or highly vegetated
– Changed to “highly productive”
Line 21: remove extremely
– Fixed
Line 21: poorly understood
– Revised accordingly
Line 21-24: combine these sentences “We characterized vegetation communities and productivity of drained thermoskarst lake basins (khasyreys) in the continuous permafrost zone of the Western Siberian Lowland (WSL), the largest peatland in the world.”
– We combined the two sentences in accordance with your proposal.
Line 25: Productivity of khasyreys vegetation was two to nine times higher than that of surrounding tundra.
– Corrected
Line 26-27: I would move this greening part to later in the abstract – it doesn’t fit here because there is no previous mention of this idea/theory. This is a discussion point so I would put it that part of the abstract.
– Corrected
Line 27: No previous mention of successional stage. If this is part of your methodology, please mention it in the methods.
Line 28: Also no previous mention of the active layer being monitored or considered in the study – please also mention this briefly in the outline of the methods.
– Fixed. We detailed the research methodology.
Line 31: change to “and nutrients are leached…”
– Corrected
Line 36: remove “on the soil surface”
– Corrected
Line 37: change to: Vascular plants, which cover between 10% and 60% of the area, are dominated by dwarf shrubs and the abundance of herbaceous species is minimal. Mosses and lichens have continuous coverage.
– Corrected
Line 38: tell us again which stage it is (late successional)
– Corrected
Line 41: remove “soil surface and microtopography of the lake bottom” this adds confusion, just having accumulation of peat is enough.
– Corrected
Line 50: change to “an increase in soil nutrient concentrations and precipitation”
– Corrected
Line 51: change to “… an increase in primary productivity…”
– Corrected
Line 52: I would say. “Increases in tundra productivity contributes to an effect known as tundra greening…” not all greening is an increase in productivity. You also need citations here and for the browning sentence (e.g. Frost et al., (2019) Tundra Greenness [In Arctic Report Card 2019]).
– Changed to: “The effect of increased productivity is known as tundra greening and has been confirmed both by satellite-derived Normalized Difference Vegetation Index (NDVI) and in-situ research [6,7].”. We have also added a citation [Frost et al., 2019].
Line 55: what effects? Be specific and mention again greening and browning. Change “such as” to “including”
Changed to: Greening and browning are observed in various subarctic regions, although trends are strongly variable and heterogenous over space and time [9].
Line 56: change “it” to “trends”
– Corrected
Line 57: this wording is confusing, 18% of the entire territory of the northern part of the WSL. What are you talking about here? Does territory refer to a specific province of a specific size? The 18% is of the entire WSL or just a sub area? Please clarify
Line 58: what does pronounced mean? Also try to be consistent in your nomenclature, is territory really necessary? Can you not just use WSL?
– Changed to: “…only 18% of the total northern WSL area had statistically significant changes in productivity, with 8.4% increasing (greening) and 9.6% decreasing (browning) [10,11].”
Line: 59: remove “It was established that”
– Corrected
Line 62: change to “observations”
– Corrected
Line 62: This sentence is redundant and doesn’t really tell the reader anything.
– We have removed this sentence.
Line 61-65: I think this should come before the specific information about the WSL.
– We moved this information above.
Line 61-62: You already said this in Line 53
– We have removed this sentence.
Line 67-68: seasonally unfrozen layer of the permafrost
– Corrected
Line 68-69: be more specific about what changes you are referring to – climatic and ground thermal regime changes. Also the wording here needs to change, permafrost stability does not respond by increasing thermokarsting. The changes in climatic and thermal regimes lead to increased thermokarsting.
– Changed to: “The changes in climatic and thermal regimes lead to increased frequency of thermokarst events…”
Line 71: This description of how thermkarst lakes drain needs to be improved. Define trough, water channel and provide a more mechanistic description of the process. This is not sufficient
Changed to: “Increasing the lake size has a positive effect on the trend of browning. In the WSL, lake drainage occurs due to the thawing of the permafrost and the formation of soil subsidences in the valley through which lake water flows. The soil stability decreases, which causes a deepening of the flow and the formation of erosion channels, through which the lake water is discharged [21].”
Note that here we reference to a paper which explicitly describe the drainage process of the WSL thermokarst lakes.
Line 73-74: I would add in here that sediments are relatively warm and nutrient rich. Also change forming, to “beginning primary succession…”
– Corrected
Line 74-79: move to Line 72. Talk about the trends of this phenomenon and then move on to the vegetation succession.
Changed the order of sentences in the paragraph.
Line 77-79: You should mention how increasing lake size is connected to browning trends.
Added sentence: Increasing lake size has a positive effect on the trend of browning.
Line 81: change border to interface
– Corrected
Line 81: Thermokarst activity
– Corrected
Line 82: change growing, to deepening (change this language throughout the paragraph)
– Corrected
Line 86-87: I think you should either remove this sentence about the upland and polygonal bogs or add more mechanistic detail on how this happens.
Fixed, added information about the enrichment mechanism.
Line 91: change to “major factors of tundra greening”
– Corrected
Line 93: change to “Nutrient-rich sediments favour…”. Remove “very”
– Corrected
Line 94: remove “Specifically” and change to “The drainage of thermokarst lakes…”
– Corrected
Line 96: observed by whom? Use occurs
– Corrected
Line 97: permafrost aggradation occurs differently in different parts of the basin
– Corrected
Line 97: what is permafrost development intensity? Depth? Temperature? Be specific
Changed to: The temperature of permafrost and ALT depends on the snow depth which is associated with vegetation growth.
Line 98: further explanation and a citation is needed for this sentence.
– Necessary explanation and citation added: “Permafrost aggradation occurs differently in different parts of the basin. The temperature of permafrost and ALT increases after the thickening of snow. In the low-Arctic tundra, the snow cover is thicker in micro-depression and on sites with shrubs. The coldest soils are located at micro-elevations (the highest point) [54,55].”
Line 98: what does “The latter” refer to?
– This sentence has been removed from the text.
Line 103: How? Describe how vegetation influences spectral reflectance
– We had in mind the effect on the specific values of the NDVI, which we specified.
Line 106: It would also be good to mention again why validation of NDVI trends is so important.
Added: Furthermore, comparing NDVI trends derived from satellite remote sensing with the results of ground-based observations should allow to reveal the mechanisms of greening and browning processes [7] and incorporate this knowledge into model predictions of ecosystem development under various climate change scenario and thus to be able to foresee the future changes.
Line 117: remote “the”
– Corrected
Line 137: further confusion about the study area as mentioned above
– Corrected
Line 153: do you do the radiocarbon dating or is this from other work? If from others, please cite that work here.
The radiocarbon age was measured in this study, and we added relevant information: The radiocarbon age of peat was determined by the liquid scintillation method using a Quantulus 1220 radiometer (Wallac, Finland) in the laboratory of the Tomsk Collective Use Center (Russian Academy of Sciences). Calibration of radiocarbon age was performed by the CALIB REV-7.10 program.
Line 154: define Ea1, etc.
– Corrected
Line 160: define Mi1, etc.
– Corrected
Line 181: add author names to citation [70]
– Corrected
Line 192: split this section into two: Vegetation properties and Biochemcial properties and microtopography
– Corrected
Line 216: remove “each”
– Corrected
Line 218: change to (detailed description is provided in…) add authors name not ref.
– Corrected
Line227: remove “one”
– Corrected
Line 229: define WTW
– Added: Multi 3510 IDS WTW (Weilheim, Germany)
Line 253: change to non-normal
– Corrected
Line 258: parameters refer to model outputs. Variables is the proper word to use here.
– Corrected
Line 276: Table 1. Lakes don’t have coasts, they have shores, please change this accordingly
– Corrected
Line 291: change to “throughout the entire basin”
– Corrected
Line 292: change to “on average”
– Corrected
Line 294: change encountered to present
– Corrected
Line 336: change to “which are common”
– Corrected
Line 339: change to “… like in the early…”
– Corrected
Table 3 can also be moved to appendix
The table has been moved to the appendix.
Line 508: remove “U-Test”
– Corrected
We thank Reviewer # 2 for very constructive remarks and corrections

Round 2
Reviewer 2 Report
I really want to thank the authors for their thorough and thoughtful response to my first review. The manuscript is much improved and following the minor editorial comments outlined below I am happy to suggest this research for publication in Plants. It is really valuable research.
I would still suggest a proof-read to polish some remaining English language and grammar mistakes prior to publication.
Specific comments
Line 21: remove thaw
Line 23: change to: influencing greening trends of Arctic tundra.
Line 23: change to: However, the magnitude and extent of this process…
Line 24: change to: productivity
Line 29: change to: In 48 sites within the different aged khasyerys…
Line 32: spell out 2 and 5 as you did in the previous sentence.
Line 34: change to: “pools of soil nitrogen and potassium”
Line44: change to: Ongoing climatic
Line 44: change to: greater nutrient availability and precipitation enhancing…
Line 49: change to: unvegetated
Line 50: remove: it
Line 51: remove the whole sentence “The present work…” This can come at the end of the intro
Line 54: define WSL and add that it is the largest peatland in the world to this sentence.
Line 58: change to: the WSL
Line 59: add a timeframe
Line 61: change to: has led to increased…
Line 72: change such to the
Line 92: leads a lower surface elevation
Line 101: change to:… will improve our understanding of the mechanisms…
Line 102: new sentence: This knowledge can be incorporated into model…
Line 103: delete: and thus to be able to foresee the future changes
Line 114: the purpose of this work is to provide…
Line 114-121: the tenses are off here. Use present tense.
Line 120-121: consistency in numbering use either i) and ii) or 1) and 2)
Line 125: in the continuous…
Line 126: mean annual temperature of -9.1
Line 128: remove: elevation
Line 132: change less to fewer
Line 135: define WRB
Line 157-158: change to: The drainage of Ea2 and Ea3 basins is still active and the majority of both basins are still covered by water.
Line 197: change to: khasyreys representing the three different age classes
Line 199: change to: (42 sites in total)
Line 220-225: change to: which are used as a proxy for productivity*. Following common methods, we calculated NDVImax from all available Sentinel-2A imagery of the study area within the timeframe of when field work was conducted (Figure 2g,h). NDVImax represents the maximum NDVI values within a growing season.
*there is some research suggesting that NDVI is actually not the best proxy for photosynthetic activity but rather productivity or vegetation greenness. The differences are subtle but I would use productivity, not photosynthesis here.
Line 247: define K, N, and P
Line 304: remove: in, and to
Line 537: change to: After decreased frequency…
Line 554: change to: … explained by the rapid uptake of nutrients…
Author Response
Reviewer 2
I really want to thank the authors for their thorough and thoughtful response to my first review. The manuscript is much improved and following the minor editorial comments outlined below I am happy to suggest this research for publication in Plants. It is really valuable research.
– We thank you for the brilliant work done, thanks to which our article has become much better and clearer for the reader. Thank you for your appreciation and all the thoughtful and important comments!
I would still suggest a proof-read to polish some remaining English language and grammar mistakes prior to publication.
– We understand the request of reviewer on a proof read. However, the manuscript was already subjected to extensive English corrections by the MDPI office, at the stage of submission, via payed service. Moreover, the corrections at the 3rd round of review were carefully incorporated in the text and this allowed greatly improving its quality. Finally, we naturally expect that our manuscript will receive full editorial proofing which is a part of the APC we pay the journal.
Specific comments
Line 21: remove thaw
– Corrected
Line 23: change to: influencing greening trends of Arctic tundra.
– Corrected
Line 23: change to: However, the magnitude and extent of this process…
– Corrected
Line 24: change to: productivity
– Corrected
Line 29: change to: In 48 sites within the different aged khasyerys…
– Corrected
Line 32: spell out 2 and 5 as you did in the previous sentence.
– Corrected
Line 34: change to: “pools of soil nitrogen and potassium”
– Corrected
Line44: change to: Ongoing climatic
– Corrected
Line 44: change to: greater nutrient availability and precipitation enhancing…
– Corrected
Line 49: change to: unvegetated
– Corrected
Line 50: remove: it
– Corrected
Line 51: remove the whole sentence “The present work…” This can come at the end of the intro
– Corrected
Line 54: define WSL and add that it is the largest peatland in the world to this sentence.
– Corrected
Line 58: change to: the WSL
– Corrected
Line 59: add a timeframe
– Corrected
Line 61: change to: has led to increased…
– Corrected
Line 72: change such to the
– Corrected
Line 92: leads a lower surface elevation
– Corrected
Line 101: change to:… will improve our understanding of the mechanisms…
– Corrected
Line 102: new sentence: This knowledge can be incorporated into model…
– Corrected
Line 103: delete: and thus to be able to foresee the future changes
– Corrected
Line 114: the purpose of this work is to provide…
– Corrected
Line 114-121: the tenses are off here. Use present tense.
– Corrected
Line 120-121: consistency in numbering use either i) and ii) or 1) and 2)
– Corrected
Line 125: in the continuous…
– Corrected
Line 126: mean annual temperature of -9.1
– Corrected
Line 128: remove: elevation
– Corrected
Line 132: change less to fewer
– Corrected
Line 135: define WRB
– Corrected
Line 157-158: change to: The drainage of Ea2 and Ea3 basins is still active and the majority of both basins are still covered by water.
– Corrected
Line 197: change to: khasyreys representing the three different age classes
– Corrected
Line 199: change to: (42 sites in total)
– Corrected
Line 220-225: change to: which are used as a proxy for productivity*. Following common methods, we calculated NDVImax from all available Sentinel-2A imagery of the study area within the timeframe of when field work was conducted (Figure 2g,h). NDVImax represents the maximum NDVI values within a growing season.
– Corrected
Line 247: define K, N, and P
– Corrected
Line 304: remove: in, and to
– Corrected
Line 537: change to: After decreased frequency…
– Corrected
Line 554: change to: … explained by the rapid uptake of nutrients…
– Corrected
